# Cryo-EM structures reveal that RFC recognizes both the 3′- and 5′-DNA ends to load PCNA onto gaps for DNA repair

Fengwei Zheng[1], Roxana Georgescu[2], Nina Y Yao[2], Huilin Li[1]*, Michael E O'Donnell[2,3]*

[1]Department of Structural Biology, Van Andel Institute, Grand Rapids, United States; [2]DNA Replication Laboratory, The Rockefeller University, New York, United States; [3]Howard Hughes Medical Institute, The Rockefeller University, New York, United States

**Abstract** RFC uses ATP to assemble PCNA onto primed sites for replicative DNA polymerases δ and ε. The RFC pentamer forms a central chamber that binds 3′ ss/ds DNA junctions to load PCNA onto DNA during replication. We show here five structures that identify a second DNA binding site in RFC that binds a 5′ duplex. This 5′ DNA site is located between the N-terminal BRCT domain and AAA+ module of the large Rfc1 subunit. Our structures reveal ideal binding to a 7-nt gap, which includes 2 bp unwound by the clamp loader. Biochemical studies show enhanced binding to 5 and 10 nt gaps, consistent with the structural results. Because both 3′ and 5′ ends are present at a ssDNA gap, we propose that the 5′ site facilitates RFC's PCNA loading activity at a DNA damage-induced gap to recruit gap-filling polymerases. These findings are consistent with genetic studies showing that base excision repair of gaps greater than 1 base requires PCNA and involves the 5′ DNA binding domain of Rfc1. We further observe that a 5′ end facilitates PCNA loading at an RPA coated 30-nt gap, suggesting a potential role of the RFC 5′-DNA site in lagging strand DNA synthesis.

*For correspondence:
Huilin.Li@vai.org (HL);
odonnel@rockefeller.edu
(MEO'D)

## Editor's evaluation

The role of Replication Factor C (RFC) in DNA replication and repair has been known for many years. RFC/PCNA binds to a double strand-single strand DNA junction with a 3'-recessed end, with the DNA passing through a central chamber in the five-subunit protein. The current paper reports structures of RFC/PCNA with two separate DNA molecules, one containing the well characterized 3'-recessed DNA and surprisingly, a second 5'-recessed DNA outside the central chamber. The paper is an important addition to understanding RFC function, particularly in DNA repair.

## Introduction

All three domains of life possess a DNA clamp and clamp loader that are essential for cell growth and proliferation (*Bell and Labib, 2016*; *Yao and O'Donnell, 2016*). The clamp and clamp loader function in genomic DNA replication, DNA damage repair, and other DNA metabolic processes (*Burgers and Kunkel, 2017*; *Harrison and Haber, 2006*; *Ohashi and Tsurimoto, 2017*; *Sancar et al., 2004*; *Su, 2006*; *Waga and Stillman, 1998*). The eukaryotic PCNA clamp and its clamp loader complex are key components of the replisome (*Ellison and Stillman, 2001*; *Hedglin et al., 2013*; *Hingorani and O'Donnell, 2000*; *Indiani and O'Donnell, 2006*; *Yao and O'Donnell, 2012*; *Yao and O'Don-nell, 2016*; *Zhang and O'Donnell, 2016*). PCNA needs to be cracked open by a clamp loader to encircle the genomic DNA because it is a topologically closed ring. The eukaryotic clamp loader is

Replication Factor C (RFC), composed of Rfc1-5, belonging to a large family of proteins involved in diverse cellular activities, referred to as the AAA+ (ATPases Associated with various cellular Activities) family (*Figure 1a*; *Cullmann et al., 1995*; *Davey et al., 2002*; *Miller and Enemark, 2016*; *Snider et al., 2008*). The ATP sites of AAA + oligomers, many of which are used in DNA replication, are located at subunit interfaces (*Erzberger and Berger, 2006*), including the bacterial and eukaryotic RFC clamp loaders (*Bowman et al., 2004*; *Jeruzalmi et al., 2001*). Interfacial ATP sites makes possible the coordination among subunits during hydrolysis and, in fact, recent deep mutagenesis of a clamp loader pentamer reveals a communication network among subunits needed to achieve the clamp loading reaction (*Subramanian et al., 2021*).

Structural studies on clamp loaders of *E. coli*, T4 phage, yeast, and human have revealed a conserved architecture of a five-subunit spiral with two tiers (*Bowman et al., 2004*; *Gaubitz et al., 2020*; *Kelch et al., 2011*; *Simonetta et al., 2009*). The upper tier is a sealed ring consisting of five collar domains - one from each subunit, whereas the lower tier is composed of five AAA+ modules and is an open spiral with a gate between the first and the fifth subunits. However, the first subunit of the eukaryotic and T4 phage clamp loaders contain an additional A' domain at their respective C termini that contacts the fifth subunit and closes the spiral gap (*Kelch et al., 2012b*). The five ATP binding sites are located at the interfaces between adjacent subunits, with one subunit contributing the Walker A (P-loop) and Walker B (DExx box, where the x can be any residue) motifs, and the other contributing the SRC motif that harbors an essential 'arginine finger' residue (*Johnson et al., 2006*). All studies thus far have revealed that primer/template (P/T) DNA with a 3'-recessed end binds inside the chamber of the ATP-bound clamp loader (*Figure 1b*; *Bowman et al., 2005*; *Kelch et al., 2011*). Moreover, clamp loaders of bacterial and eukaryotic systems only need ATP binding to crack open the clamp ring; neither ATP hydrolysis nor DNA is required (*Hingorani and O'Donnell, 1998*; *Kelch et al., 2012b*; *Kelch et al., 2012a*; *Turner et al., 1999*; *Zhuang et al., 2006*). DNA binding appears to properly engage the ATPase sites for hydrolysis, leading to closure of the clamp around the duplex and dissociation of the clamp loader from DNA (*Anderson et al., 2009*; *Chen et al., 2009*; *Gaubitz et al., 2022*; *Kelch et al., 2012b*; *Liu et al., 2017*; *Marzahn et al., 2015*; *Sakato et al., 2012a*; *Trakselis et al., 2003*).

The replication clamps are ring-shaped complexes that encircle dsDNA and slide freely on the duplex, conferring processivity to their respective replicative DNA polymerase (*Kunkel and Burgers, 2014*; *Kunkel and Burgers, 2017*; *Nick McElhinny et al., 2008*; *ODonnell and Kurth, 2013*; *Pursell et al., 2007*; *Yao and O'Donnell, 2021*). In eukaryotes, PCNA (Proliferating Cell Nuclear Antigen) functions with replicative polymerases Pol δ on the lagging strand and Pol ε on the leading stand (*Kunkel and Burgers, 2014*; *Kunkel and Burgers, 2017*; *Nick McElhinny et al., 2008*; *ODonnell and Kurth, 2013*; *Pursell et al., 2007*; *Yao and O'Donnell, 2021*). Structural studies showed that, the *E. coli* β-clamp is a homodimer, while the T4 phage clamp and archaeal and eukaryotic PCNA clamps are homotrimers, all of which display six domains having similar chain folds and remarkable pseudo sixfold symmetric architecture with an obvious ancestor in evolution (*Gulbis et al., 1996*; *Kong et al., 1992*; *Krishna et al., 1994*; *Matsumiya et al., 2001*; *Yao and O'Donnell, 2016*). The monomers are arranged in head-to-tail manner, with 12 α-helices that line the inner cavity and a continuous layer of antiparallel β-sheet forming the outside perimeter, even at the protomer interfaces. In each case, the domains within each monomer are linked by an interdomain connecting loop (IDCL). The diameter of the PCNA inner lumen is about 38 Å, and the inner surface is also basic, thus suitable to encircle a 20 Å wide dsDNA and is shown to slide on DNA freely (*Bloom, 2009*; *Li et al., 2021*). PCNA interacts with numerous proteins in replication, repair, and cell cycle control (*Kelman, 1997*; *Maga and Hubscher, 2003*). All replication clamps characterized so far mainly interact with their partners by their respective carboxy-terminal 'front' faces. The 'front' face of each monomer possesses a hydrophobic pocket that interacts with the PIP motif (PCNA-interacting peptide), or modified PIP motifs of clamp interactive proteins (*Altieri and Kelman, 2018*; *Fernandez-Leiro et al., 2015*; *Gulbis et al., 1996*; *Lancey et al., 2020*; *Madru et al., 2020*; *Prestel et al., 2019*; *Zheng et al., 2020*).

A recent cryo-EM study captured yeast RFC in a series of intermediate states during loading of the PCNA clamp onto DNA, including an intermediate in which the PCNA ring is open by 20 Å, sufficiently wide for dsDNA to pass through (*Gaubitz et al., 2022*). This conformation was not observed in the crystallographic study of the T4 phage clamp−loader−DNA complex (*Kelch et al., 2011*). The new structure confirms that clamp opening does not need ATP hydrolysis. However, PCNA closure around

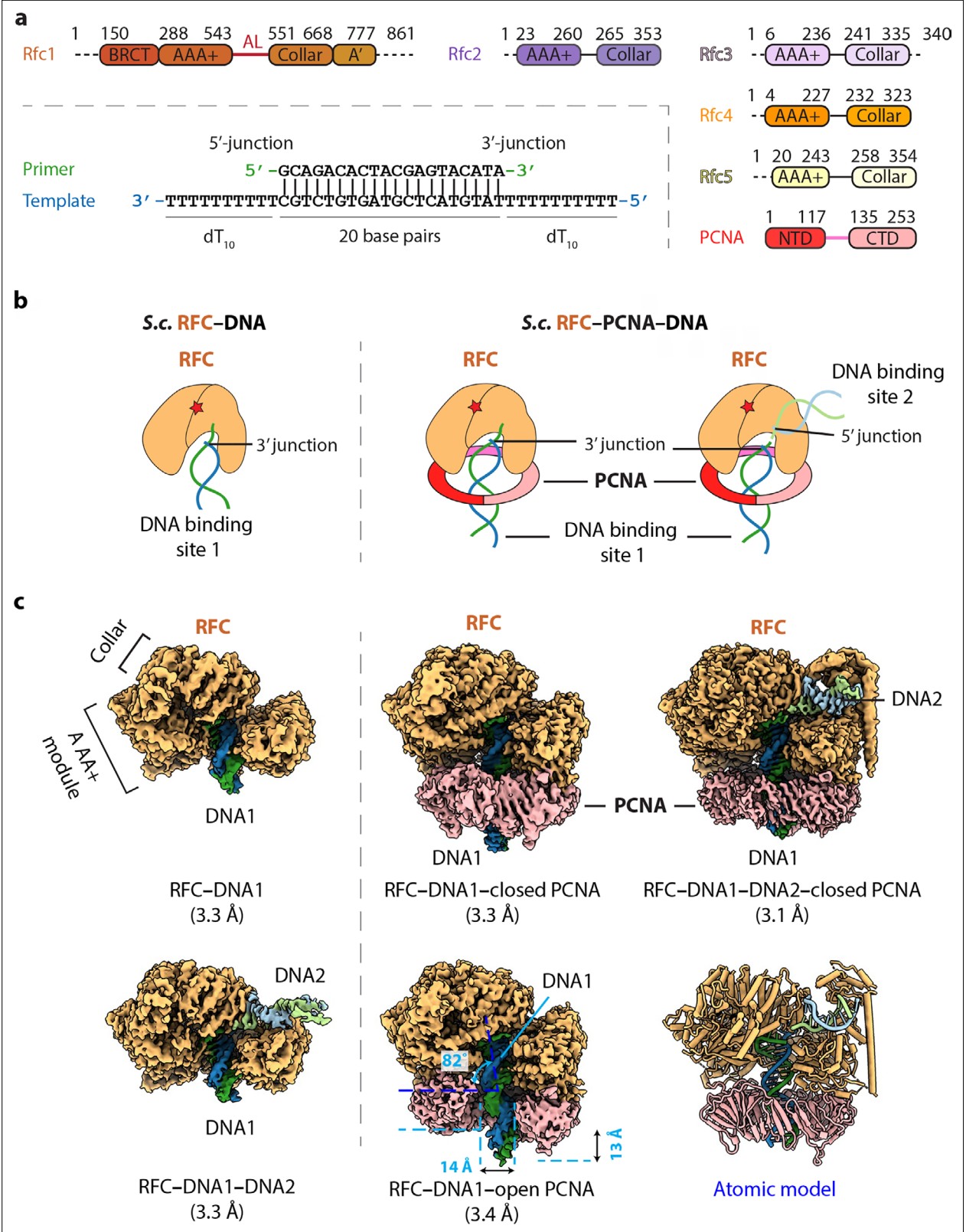

**Figure 1.** Cryo-EM captured multiple S.c. RFC–(PCNA)–DNA complexes. (**a**) Domain architecture of RFC, PCNA, and the template/primer DNA. Dashed lines indicate unsolved regions. AL is the alternative linker between Rfc1 AAA+ and collar domain. Pink line between PCNA NTD and CTD represents the inter-domain connecting loop (IDCL). DNA used in this study harbors both 5'-DNA and 3'-DNA junctions. (**b**) Sketch illustrating the previously established DNA1 binding site recognizing the 3'-DNA junction in the RFC central chamber (left) and the novel 5'-DNA2 site recognizing the 5'-DNA

*Figure 1 continued on next page*

*Figure 1 continued*

junction on Rfc1 shoulder. (**c**) 3D maps of the five loading intermediates: RFC–3'-DNA1 and RFC–3'-DNA1–5'-DNA2 (left), RFC–3'-DNA1–PCNA in closed and open spiral form (middle), and RFC–3'-DNA1–5'-DNA2–PCNA (right). These maps are displayed at the same threshold (0.2). RFC is in orange, PCNA pink. DNA strands are individually colored.

The online version of this article includes the following figure supplement(s) for figure 1:

**Figure supplement 1.** Cryo-EM of in vitro assembled *S. cerevisiae* RFC–PCNA–DNA complexes.

**Figure supplement 2.** Cryo-EM image processing workflow.

**Figure supplement 3.** Resolution estimation of the six 3D EM maps of the RFC loading complexes.

**Figure supplement 4.** Comparison of the PCNA poses in various RFC−(DNA)−PCNA structures.

dsDNA induced conformation changes that tighten up the RFC ATP binding sites at subunit interfaces and likely stimulate the ATPase. ATP hydrolysis presumably releases RFC from PCNA, making PCNA available to bind its functional partners, such as Pols δ and ε for DNA synthesis, and FEN1 and ligase for Okazaki fragment maturation (*Bell and Labib, 2016*; *Yao and O'Donnell, 2016*).

Rad24-RFC is an alternative clamp loader in which Rad24 replaces Rfc1 of the RFC clamp loader and loads the DNA damage signaling 9-1-1 clamp onto a 5'-recessed DNA end (*Ellison and Stillman, 2003*; *Majka et al., 2006*; *Majka and Burgers, 2003*). We and others recently determined the cryo-EM structures of the Rad24-RFC−9-1-1 clamp complex bound to a 5'-ss/ds DNA (*Castaneda et al., 2022*; *Zheng et al., 2022*). We observe two structures in which the 9-1-1 clamp is either closed or had a 27 Å opening. These structures revealed that the 5'-junction DNA is held specifically by the Rad24 subunit in a long basic groove between the AAA+ module and collar domain, and that the 3' ssDNA overhang enters the inner chamber of the alternative loader. This DNA interaction mode is unexpected, as it is entirely different from the DNA binding pattern of RFC which binds the 3'-recessed dsDNA inside the central chamber, while the 5' ss/ds DNA is not observed to bind the clamp loader as reported in the T4 phage and yeast structures (*Kelch et al., 2011*).

The surprising location of the 5'-ss/ds DNA binding site on the Rad24 subunit, but outside the central chamber of Rad24-RFC, led us to ask if RFC also contains a second DNA binding site that binds 5' ss/ds DNA above the clamp ring. If true, the 5' ss/ds DNA binding site may explain the previously reported 5'-ss/ds DNA junction binding by the human RFC1 BRCT domain (*Kobayashi et al., 2010*; *Kobayashi et al., 2006*). In addition, dual binding of 3' and 5' recessed DNA ends by RFC suggests that RFC may bind both ends of a DNA gap, congruent with genetic studies demonstrating that mutants in the N-terminal region of Rfc1 are defective in excision gap repair (*Aboussekhra et al., 1995*; *Gomes et al., 2000*; *Li et al., 1994*; *McAlear et al., 1996*; *Shivji et al., 1992*).

These clues to dual DNA binding sites in RFC have inspired us to examine the structure of *S. cerevisiae* RFC in the presence of a double-tailed DNA in which both the 3'-recessed and 5'-recessed ends are present on the same molecule. We used this strategy in our previous Rad24-RFC study that revealed 5' ss/ds end binding to Rad24 (*Zheng et al., 2022*). Indeed, we found that RFC can bind up to two DNA molecules at the same time, one being the well-established site wherein a recessed 3' ss/ds DNA binds inside the pentamer central chamber, and a second site that binds a recessed 5' ss/ds DNA at the Rfc1 "shoulder" region outside the central chamber. This surprising dual DNA site binding mode of RFC described here resembles the DNA binding by Rad24-RFC (*Castaneda et al., 2022*; *Zheng et al., 2022*). We also examine the affinity and PCNA clamp loading efficiency of RFC at different sized DNA gaps. The results reveal enhanced RFC binding at 5–50 nt gaps, and 5' end stimulation in loading of PCNA at a 10-nt gap and 30-nt gap, consistent with a role of RFC 5' end binding in gap repair. The results also show a measurable enhancement in RFC loading of PCNA at an RPA-coated 30-nt gap, suggesting that RFC may act in longer excision tracks, and possibly binds both 3' and 5' ends of an Okazaki fragment fill-in site, perhaps to protect the 5' end at Okazaki fragments and prevent the DNA damage repair pathway triggered by it.

## Results

## RFC can engage a 3′-DNA in the central chamber simultaneously with a 5′-DNA located on the Rfc1 'shoulder'

The *S. cerevisiae* PCNA clamp and RFC clamp loader were separately expressed and purified from *E. coli* (see Materials and Methods). We assembled the loading complexes in vitro by directly mixing the purified proteins with a double-tailed primer/template DNA with both 3′-recessed and 5′-recessed ends (*Figure 1a*) in the presence of 0.5 mM slowly hydrolysable ATP analog ATPγS (see details in Materials and Methods). Use of DNA having both 3′ and 5′ ssDNA ends strategically enables the

**Table 1.** Cryo-EM data collection, refinement, and atomic model validation.

| Structures | RFC−3′-DNA1 | RFC−3′-DNA1−5′-DNA2 | RFC−3′-DNA1−PCNA (open) | RFC−3′-DNA1−PCNA (closed) | RFC−3′-DNA1−5′-DNA2−PCNA |
|---|---|---|---|---|---|
| **Data collection and processing** | | | | | |
| Magnification | 105,000 | | | | |
| Voltage (kV) | 300 | | | | |
| Electron dose (e⁻/Å²) | 66 | | | | |
| Under-focus range (μm) | 1.3–1.9 | | | | |
| Pixel size (Å) | 0.828 | | | | |
| Symmetry imposed | C1 | | | | |
| Initial particle images (no.) | 1,039,425 | | 908,744 | | |
| Final particle images (no.) | 334,876 | 315,619 | 118,384 | 166,348 | 432,904 |
| Map resolution (Å) | 3.33 | 3.25 | 3.41 | 3.30 | 3.09 |
| FSC threshold | 0.143 | | | | |
| Map resolution range (Å) | 2.0–12.0 | 2.0–13.0 | 2.0–12.0 | 2.0–12.0 | 2.1–13.0 |
| **Refinement** | | | | | |
| Initial model used (PDB code) | RFC−3′-DNA1−5′-DNA2−PCNA of this study | | | | 1SXJ, 7SGZ, 2K6G |
| Map sharpening B factor (Å²) | −93.8 | −100.8 | −97.3 | −118.6 | −116.6 |
| Map to model CC$_{mask}$ | 0.70 | 0.69 | 0.82 | 0.81 | 0.82 |
| **Model composition** | | | | | |
| Non-hydrogen atoms | 13,690 | 14,535 | 20,573 | 20,607 | 22,831 |
| Protein and DNA residues | 1,653; 24 | 1,701; 45 | 2,480; 42 | 2,486; 42 | 2,717; 63 |
| Ligands | 9 | 9 | 9 | 9 | 9 |
| **R.m.s. deviations** | | | | | |
| Bond lengths (Å) | 0.003 | 0.003 | 0.003 | 0.003 | 0.003 |
| Bond angles (°) | 0.616 | 0.634 | 0.596 | 0.623 | 0.562 |
| **Validation** | | | | | |
| MolProbity score | 1.72 | 1.72 | 1.66 | 1.56 | 1.57 |
| Clashscore | 9.60 | 10.00 | 8.57 | 7.59 | 7.57 |
| Poor rotamers (%) | 0.14 | 0.13 | 0.05 | 0.09 | 0.04 |
| **Ramachandran plot** | | | | | |
| Favored (%) | 96.64 | 96.78 | 96.83 | 97.20 | 97.14 |
| Allowed (%) | 3.36 | 3.22 | 3.17 | 2.80 | 2.86 |
| Disallowed (%) | 0 | 0 | 0 | 0 | 0 |

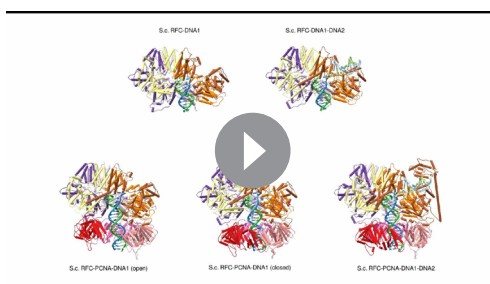

**Video 1.** A side-by-side comparison of the five structures obtained in this study. The five atomic models are rotated around the Y axis by 360° followed by another 360° rotation around the X axis. These models are first aligned to the RFC–3'-DNA1–5'-DNA2–PCNA model (lower right) by the subunits Rfc1 and Rfc4, and shown separately as cartoons in the same color scheme as in Figure 3a.
https://elifesciences.org/articles/77469/figures#video1

unbiased binding of either end to sites on RFC and 2D classification of cryo-EM images demonstrated the assembly of a variety of loading intermediates, including RFC bound to either one or two DNA molecules, and with the loader either engaged or not with PCNA (*Figure 1—figure supplement 1*). Subsequent 3D classification and 3D variability analysis (3DVA) led to the capture of five 3D maps: (1) RFC bound to 3' recessed DNA in the central chamber (termed 3'-DNA1 herein) at 3.3 Å resolution, (2) RFC bound to two DNA molecules at 3.3 Å resolution, with 3'-recessed DNA1 in the central chamber plus a 5' ss/ds DNA bound to the Rfc1 shoulder (referred to here as 5'-DNA2) (*Table 1*). The 5'-DNA2 is nearly parallel with and above the position that is occupied by the PCNA ring, (3) RFC-closed planar PCNA with 3'-DNA1 bound in the central chamber and threaded through the PCNA ring at 3.3 Å resolution, (4) RFC-open PCNA with 3'-DNA1 bound in the central chamber and threaded through an open PCNA ring that assumes a right-hand spiral with a 14 Å opening in the clamp at 3.4 Å resolution, and (5) RFC-closed PCNA with 3'-DNA1 in the central chamber plus a molecule of 5'-DNA2 on the Rfc1 shoulder at 3.1 Å resolution (*Figure 1c*, *Figure 1—figure supplements 2–3*). The fifth map is a composite map produced by combining the 3.1 Å map of RFC−3'-DNA1−5'-DNA2−PCNA with a locally refined map focusing around Rfc1 NTD (including the BRCT domain and AAA+ module) and 5'-DNA2.

Interestingly, the RFC structure is highly similar among the five structures captured here, with main chain Cα RMSDs (root-mean-square deviation) ranging from 0.6 Å to 1.0 Å (*Figure 1—figure supplement 4*, *Video 1*). In all five structures, the Rfc1 AAA+ module sits above the PCNA protomer (when PCNA is present) at the right side of the DNA entry gate (viewing down from the C-terminal collar), while the Rfc5 AAA+ module is above the PCNA protomer at the left side of the DNA entry gate (*Figure 1—figure supplement 4*). These structural features are consistent with previous studies (*Bowman et al., 2004*; *Gaubitz et al., 2020*; *Kelch et al., 2011*). The closed-ring PCNA structure of RFC−3'-DNA1−PCNA is comparable with the RFC−PCNA structures determined in the absence of DNA (*Bowman et al., 2004*; *Gaubitz et al., 2020*; *Figure 1—figure supplement 4*). However, DNA makes the structure significantly more compact, as PCNA tilts up by 10° toward RFC, forming a tighter interface (*Figure 1—figure supplement 4*). In the open PCNA structure, Rfc1 and Rfc5 each bind one different PCNA protomer, holding the two protomers 14 Å apart in-plane and 13 Å apart out-of-plane (*Figure 1c*, middle panel). Because the 14 Å gap in PCNA is narrower than the 20 Å width of a dsDNA, and 3'-DNA1 is already inside the RFC central chamber, this is likely a post-entry intermediate in which the DNA gate has partially closed. These open and closed PCNA states resemble the open and closed clamp states observed in the T4 phage clamp-clamp loader complex (*Kelch et al., 2011*) and the recent RFC−PCNA−DNA structure (*Gaubitz et al., 2022*; *Figure 1—figure supplement 4c*).

While RFC has been studied for many years, and is known to bind recessed 3'-DNA in the central chamber, it has not previously been recognized to also contain a recessed 5'-DNA binding site. Indeed, earlier studies were performed on RFC in which the N-terminal region of Rfc1 containing part of the 5'-DNA site was deleted (*Gomes et al., 2000*; *McAlear et al., 1996*; *Xie et al., 1999*), It is well known that RFC binds PCNA in an ATP (ATPγS) dependent fashion (*Zhuang et al., 2006*), similar to earlier studies of the *E. coli* clamp-clamp loader interaction that is dependent on non-hydrolysable nucleoside triphosphate (*Turner et al., 1999*). We also show here that RFC interacts with DNA either in the absence of PCNA (RFC−3'-DNA1 and RFC−3'-DNA1+5'-DNA2), or in the presence of PCNA (RFC−3'-DNA1−open spiral PCNA, and RFC−3' DNA1−closed ring PCNA, and RFC−3'-DNA1+5'-DNA2−PCNA) (*Figures 1c and 2a–d*). This observation suggests that RFC can interact with

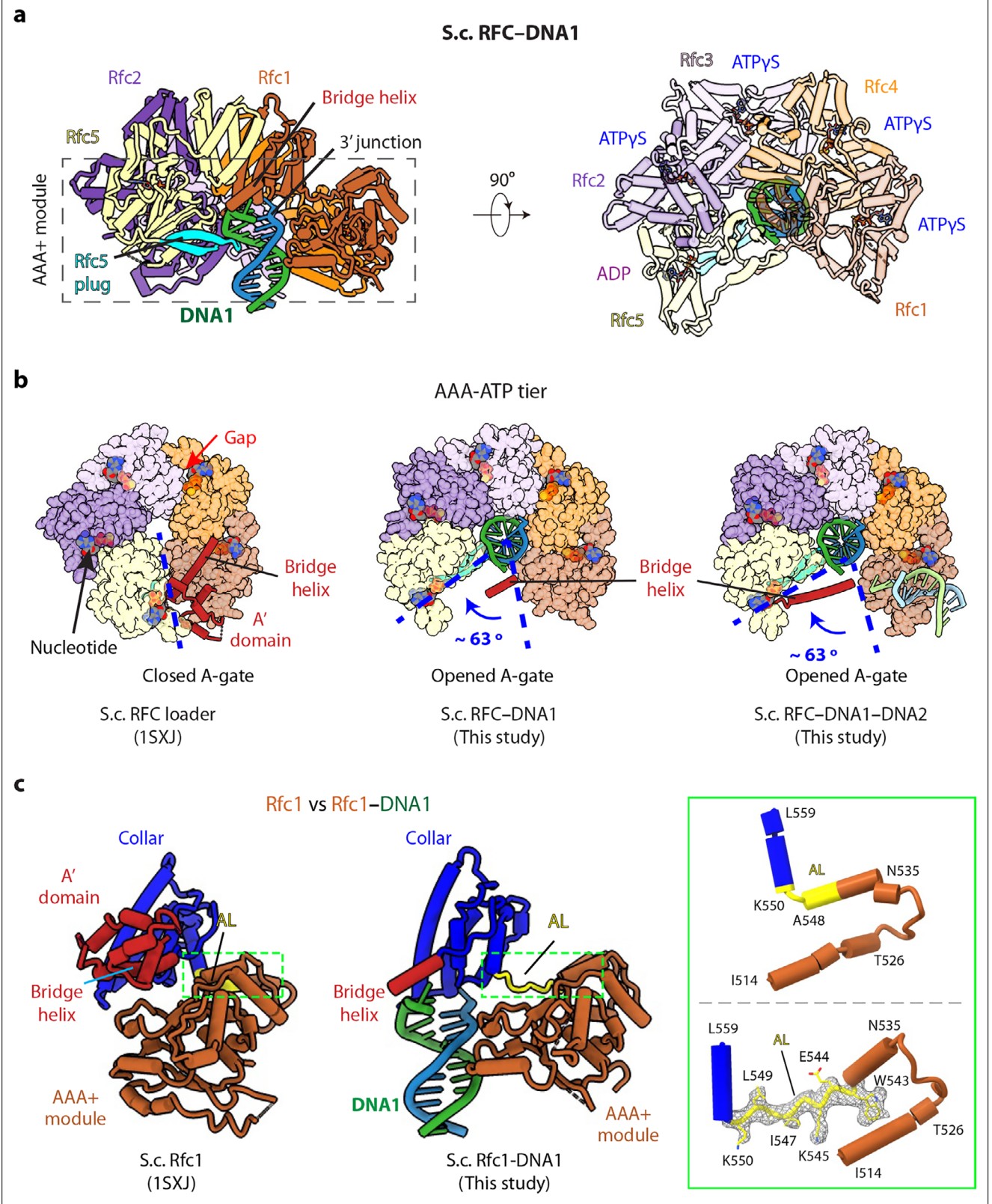

**Figure 2.** 3'-DNA1 binding induces conformation changes and formation of the 5'-DNA2 site. (**a**) Structure of RFC–3'-DNA1 complex colored by individual subunits and DNA strands (template in green and primer in steel blue) in a side (left) and a top view (right). The Rfc5 plug inserted into the DNA major groove is cyan. The four ATPγS and one ADP (only at the Rfc5:1 interface) are in sticks. (**b**) Comparison of the five AAA+ domains of RFC in the RFC–PCNA structure in the absence of DNA (left) or bound to 3'-DNA1 (middle) or bound to 3'-DNA1 and 5'-DNA2 (right) (the AAA-Lid domains

*Figure 2 continued on next page*

*Figure 2 continued*

are omitted for clarity). (**c**) Rfc1 in the absence of DNA (PDB entry 1SXJ) is superimposed with the AAA+ module with Rfc1 in the presence of 3′-DNA1 (this study), revealing large scale leftward movements of the collar and A′ domains (along with the 3′-DNA1). The region in the dashed green box is enlarged in the right panels showing the conformational changes of the alternative linker (AL), the EM density of the changed AL is shown in meshes. The structures in panels b-c are aligned by superimposing Rfc1 and Rfc4.

The online version of this article includes the following figure supplement(s) for figure 2:

**Figure supplement 1.** Structural comparison of the RFC loaders in different functional states.

**Figure supplement 2.** 3′-DNA1 binding induces major conformation changes in Rfc1 to form the 5′-DNA2-binding groove.

DNA independent of PCNA, which we presume reflects the stability of this structure using ATPγS, compared to freely hydrolysable ATP as determined by earlier studies (*Gomes and Burgers, 2001*).

## 3′-DNA1 binding induced large conformational changes in RFC that facilitate the 5′-DNA2 binding site

RFC bound to 3′-DNA1 is in the shape of a two-tiered right-handed spiral with a closed collar ring on the top and an open spiral AAA+ tier on the bottom (*Figure 2a–b*). The five RFC subunits are arranged counterclockwise from Rfc1, Rfc4, Rfc3, Rfc2, and Rfc5 when viewed from the top (i.e., from the C-terminal collar domain). Among the five nucleotide binding sites, four were occupied by ATPγS and one by ADP (between Rfc5 and Rfc1), illustrated in *Figure 2a*, viewed from the C-terminal domains and upon removal of the C-terminal collar domains. ADP is possibly an ATPγS hydrolysis product by Rfc5. The Rfc5 subunit contains a 'β-hairpin plug' element that protrudes into the central chamber. In the absence of 3′-DNA1, the Rfc5 plug is largely disordered, and there is a gap at the ATP binding interface between Rfc4 and Rfc3 in the RFC−PCNA structures in the presence of ATPγS and Mg$^{2+}$ (*Bowman et al., 2004*; *Gaubitz et al., 2020*; *Figure 2b*). In the 3′-DNA1- or 3′-DNA1−5′-DNA2-bound struc-

tures, the Rfc5 β-hairpin plug becomes ordered and inserts into the major groove of 3′-DNA1 located in the central chamber (*Figure 2a*, *Figure 2—figure supplement 1*). Furthermore, the AAA-ATP domains of Rfc3, 2 and 5 undergo a large rotation of 63° to open a gate, referred to as the A-gate (*Gaubitz et al., 2022*), between the AAA+ module and the A′ domain of Rfc1 and to tighten up the gapped ATP-binding interface between Rfc4 and Rfc3, perhaps stimulating the ATPase activity at this site (*Figure 2b*). However, the Rfc1 A′ domain becomes largely flexible upon 3′-DNA1 binding, and only the bridge helix – the α-helix connecting the collar and A′ domains (aa 669–689) – remained stable (*Figure 2a–b*).

The 3′-DNA1 binding site in the chamber is largely pre-formed in RFC, but the 5′-DNA2 site, located between the collar domain and the AAA+ module of Rfc1, is not formed in the absence of 3′-DNA1 binding, but it is formed upon binding 3′ DNA. Specifically, the Rfc1 collar region is very close to its AAA+ module with only a single residue (Leu-549) linking these two domains (*Figure 2c*). Our structures reveal that, concomitant to the large-scale rotation of the Rfc3-2-5 half ring upon binding 3′-DNA1, as described above, 3′-DNA1 binding also causes conformation changes in the last α-helix (Asn-535 to Ala-548) of the Rfc1 AAA+ module and the first helix of the collar domain. Specifically, in Rfc1, the second

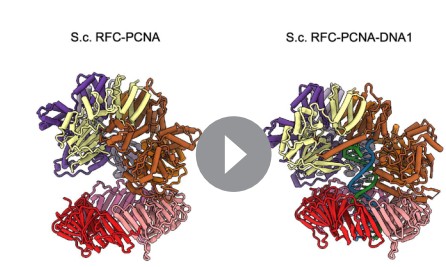

S.c. RFC-PCNA          S.c. RFC-PCNA-DNA1

**Video 2.** Morph of Rfc1 from in the RFC−PCNA structure to in the structure of RFC−3′-DNA1−PCNA. The two models are aligned by the subunits Rfc1 and Rfc4, and shown first side by side in cartoon view. Then, the RFC−PCNA is changed to pale green, and RFC−PCNA−3′-DNA1 to rosy brown with DNA1 in cyan. Next, the two structures are superimposed and the PCNA clamp and DNA are removed for clarity. The remaining structures rotate around the X axis by 35° for better visualization of the 5′-DNA2 binding groove. The subunits Rfc3, 2 and 5, along with the entire collar region, move towards lower left to open the gate between the AAA+ module and the A′ domain of Rfc1 allowing DNA1 entry into the RFC chamber (step 1). Concomitantly, a large groove is introduced between Rfc1 collar domain and AAA+ module to accommodate 5′-DNA2 at its shoulder (step 2). The Rfc1 BRCT domain participating in the shoulder 5′-DNA2 binding is also shown.

https://elifesciences.org/articles/77469/figures#video2

half of the last AAA+ helix (Trp-543 to Ala-548) and the first residue of the collar helix (Lys-550) unwind and they, together with Leu-549, become an extended loop. This loop allowed the counter-clockwise movement of the Rfc1 collar and A′ domains, leading to the formation of the 5′-DNA2-binding cleft (*Figure 2c*, *Figure 2—figure supplement 2*, *Video 2*). We have termed the loop that forms upon binding 3′-DNA1 as the 'alternative linker' (AL), because it is reminiscent of the Rad24 long linker between the AAA+ module and collar domain which allows the DNA binding at the Rad24 shoulder (*Figure 1—figure supplement 4d*). Importantly, a recent structural study revealed a similar RFC form induced by ATPγS binding without DNA (*Gaubitz et al., 2022*).

Among the eight 3D subclasses of RFC−DNA derived from 3D variability analysis (3DVA), two subclasses had either 3′-DNA1 alone or had both 3′-DNA1 and 5′-DNA2 sites occupied *Figure 1—figure supplement 2*; each type accounted for about 31% of the ~1 million particle population. The remaining four subclasses had 3′-DNA1 and partially occupied 5′-DNA2 sites, and they account for 38% of the population. However, we did not observe an RFC−5′-DNA2 structure in which the 5′-DNA2 site is occupied, and the 3′-DNA1 site is empty. Based on our structural analysis, we suggest that RFC binds 3′-DNA1 inside the central chamber first, and that 3′-DNA1 binding induces the above-described conformation changes that configure the 5′-DNA2 binding site.

## The Rfc1 BRCT domain assists 5′-DNA2 binding

In the RFC−3′-DNA1−5′-DNA2−PCNA structure, 5′-DNA2 binding stabilized the otherwise flexible Rfc1 BRCA1 C-terminal homology (BRCT) domain (*Figure 3a–b*). The yeast Rfc1 BRCT domain resembles a clenched hand on an arm and is composed of a core subdomain (hand) followed by a vertical long α-helix (arm) (*Figure 3a*). The core adopts the canonical BRCT domain folding and is similar to the NMR structure of the human RFC1 BRCT domain (*Kobayashi et al., 2010*; *Figure 3b*). The BRCT core contains a 4-stranded (β1–4) β-sheet flanked by three α-helices, with α1 and α3 at one side and α2 at the other side, and an additional α1′-helix at the N-terminus. The loop connecting α1′ and β1 in our yeast Rfc1 BRCT structure (Rfc1 aa 113–161; 49 aa) is twice the length of the corresponding loop in the human RFC1 BRCT domain.

## RFC interactions with 3′-DNA1 and 5′-DNA2

The RFC interaction with 3′-DNA1 is similar in all five captured structures. 3′-DNA1 is recognized by a series of α-helices in the RFC−3′-DNA1−5′-DNA2 structure (*Figure 3c*). Helices α4 and α5 of the AAA-ATP domains of Rfc1-4 bind and spiral around the template 3′-DNA1 strand, similar to the T4 clamp-clamp loader system (*Kelch et al., 2011*). In particular, the α4 Ser-384 of Rfc1; α4 Ile-86 of Rfc4; α4 Ile-90, Arg-94 and α5 Thr-123 of Rfc3 H-bond with the phosphate backbone of the template strand; the α4 Ile-103 of Rfc2 and Arg-81 in the β-hairpin of Rfc5 (Rfc5 plug) also H-bond with the phosphate backbone of the template strand. Rfc4 Lys-275 and the Rfc5 plug Asn-80 H-bond with the primer strand. The α5 Arg-434 of Rfc1 inserted into the minor groove of DNA1, the separation pin, Rfc1 Trp-638, is equivalent to the separation pin observed in the bacterial β-clamp loader (*Simonetta et al., 2009*), disrupting dT-19 and dA-20 at the 3′-end of the primer from the template, and the two separated bases are stabilized by the Rfc1 Phe-582 and Rfc4 Lys-275, respectively. In addition, Rfc1 Arg-632 and Gln-636 also H-bonds with the unwound template dA-12 (*Figure 3c*). Single base unwinding by RFC at the 3′-junction DNA has been reported previously (*Gaubitz et al., 2022*).

In overview, RFC mainly contacts the template strand of 3′-DNA1, but there are a few contacts to the primer strand: specifically, the Rfc5 plug touches both strands, and Rfc1 Phe-582 and Rfc4 Lys-275 also contact the primer strand. Regardless, the main contact of RFC to the primer/template DNA is with the template strand, as it is in the *E. coli* and T4 phage clamp loader-DNA complexes (*Bowman et al., 2005*; *Kelch et al., 2011*).

Only the first 10 bp of the duplex portion of the 5′-DNA2 containing the 5′-recessed end is stably bound to RFC and modelled. 5′-DNA2 is sandwiched between the BRCT domain on the top and the Rfc1 AAA+ module on the bottom (*Figure 3a and d*). Therefore, 5′-DNA2 is primarily bound by the first clamp loader subunit, Rfc1, in a manner similar to Rad24 DNA binding in Rad24-RFC (*Figure 2—figure supplement 2c, d*; *Castaneda et al., 2022*; *Zheng et al., 2022*). However, the specific contacts are distinct. Four basic residues on the top of the Rfc1 AAA-ATP interact with the double strand region of 5′-DNA2: Lys-314 and Asn-459 H-bond with the phosphate backbone of the primer and template respectively, and Arg-476 and Arg-477 sandwich the template strand, with Arg-476 inserted in the

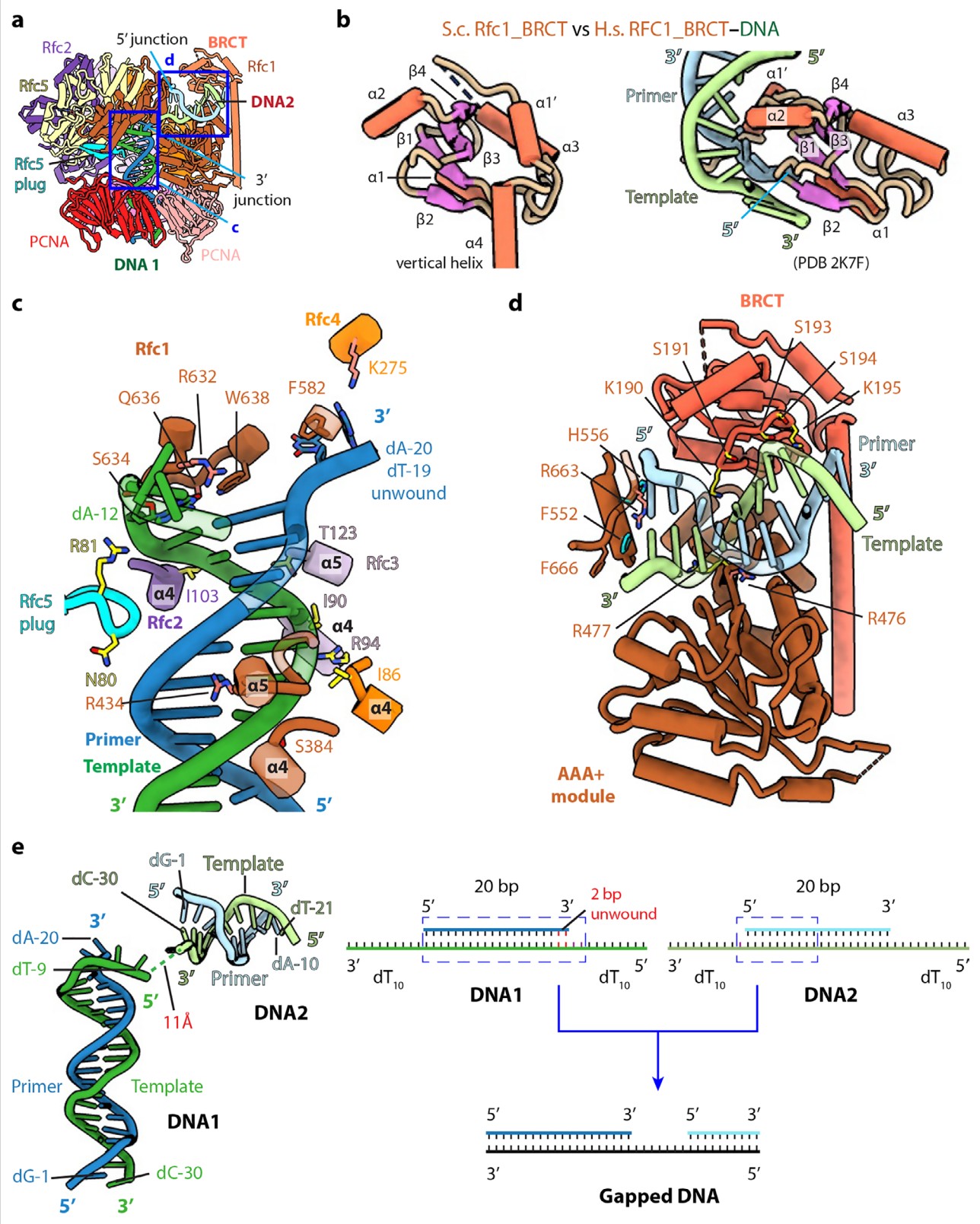

**Figure 3.** The BRCT domain of Rfc1 stabilizes 5'-DNA2 in the quaternary RFC–3'-DNA1–5'-DNA2–PCNA complex. (**a**) The atomic model of the quaternary RFC–3'-DNA1–5'-DNA2–PCNA complex. The Rfc1 BRCT domain caps 5'-DNA2. Regions in the two blue boxes are enlarged in (**c**) and (**d**). (**b**) Structure of the yeast Rfc1 BRCT domain in the context of the full RFC complex bound to DNA (DNA not shown) compared with the human RFC1 BRCT domain (aa 375–480; PDB entry 2K7F). The human RFC1 BRCT was computationally modeled to recognize the 5'-junction of dsDNA by the

*Figure 3 continued on next page*

Figure 3 continued

end region of β1–2. (**c**) Enlarged view showing interactions between RFC and 3'-DNA1. Helices α4 and α5 of the AAA-ATP domain of Rfc1, 4, 3 and 2 wrap around the template strand. DNA is shown as stubs with the separated base pair (dA-12:dT-19) and base dA-20 in primer strand shown in sticks. Residues Arg-632 and Gln-636 (from Rfc1), Lys-275 (from Rfc4) contacting the bases of DNA are in salmon sticks. Residues H-bonding with DNAs are in yellow. The main chain nitrogen atoms of Ile-86, Ile-90, and Ile-103 in the α4 helices of Rfc4, 3 and 2 form H-bonds with template DNA. (**d**) Enlarged view of the 5'-DNA2 binding region. The positively charged Rfc1 BRCT domain on top and positively charged AAA+ module from the bottom stabilize the 5'-DNA2. Key residues surrounding 5'-DNA2 are labelled. The α-helix of Rfc1 collar domain harboring Phe-552 and His-556 (cyan sticks) blocks the 5'-DNA2 5'-junction. The "separation pin" residue Phe-666 located at the DNA2 5'-junction, Arg-476 and Arg-663 (salmon sticks) insert into the DNA minor groove, while Arg-477 (yellow stick) point to the DNA major groove. For clarity, only partial protein secondary structures are shown. Unless noted otherwise, the same color and rendering scheme is used in all figures. (**e**) The arrangement of 3'-DNA1 and 5'-DNA2 in RFC (left) resembles a gapped DNA (right).

The online version of this article includes the following figure supplement(s) for figure 3:

**Figure supplement 1.** The binding cavity for the 5'-recessed end of the shoulder DNA2.

minor groove. The back surface of the 5'-DNA2-binding groove is basic, lined by a series of alkaline residues including Lys-190 and Lys-195 of the BRCT domain. The 5'-DNA2 interaction with Rfc1 BRCT in the full RFC complex as observed in our structure is very different from the previous prediction based on the NMR structure of the truncated human RFC1 BRCT domain (*Figure 3b and d*). It is currently unclear if the predicted binding mode is an in vitro artifact of using an isolated domain or might be an intermediate preceding the stable binding observed in our structure. The 5'-recessed end of the 5'-DNA2 is blocked by an α-helix (harboring two aromatic residues Phe-552 and His-556) of the collar domain, Phe-666 is precisely located at the 5'-junction, perhaps functioning as the 'separation pin' for 5'-DNA2; and Arg-663 in the minor groove appears to stabilize the 5'-junction. The recessed 5' end is in a cavity lined by three basic residues, Lys-208 of the BRCT domain and His-556 and His-659 of the Rfc1 collar domain. The cavity is larger than necessary for accommodating the 5'-OH used in this study but fits the 5'-phosphate snuggly with two potential H-bonds (*Figure 3—figure supplement 1a-b*). The cavity for the 5'-recessed end in the alternative clamp loader Rad24-RFC is very similar and is also lined by one Lys residue and two His residues (*Figure 3—figure supplement 1c*).

## 3'-DNA1 and 5'-DNA2 arrangement in RFC mimics a single strand DNA gap

Even though both 3'-DNA1 and 5'-DNA2 are identical, and both have 3' and 5' junctions, our structural analysis reveals that one DNA (3'-DNA1) engages the RFC central chamber with the 3'-recessed end, and the other DNA (5'-DNA2) is bound by Rfc1 such that the 5' end is pointed towards the central chamber of the RFC (*Figures 1b–c and 3c–d*). Surprisingly, we found that the two identical DNA molecules, when bound to RFC, resemble a single gapped DNA (*Figure 3e*). As we described above, RFC stabilizes 2 unpaired and unwound 2 additional bases in the 3'-DNA1, and Rfc1 stabilized 1 unpaired base in the 5'-DNA2. Furthermore, the 11 Å space between the 5' end of the 3'-DNA1 template strand and the 3'-end of the 5'-DNA2 template strand can accommodate 2 additional bases. Therefore, DNA1 and DNA2 are held by RFC like a gapped DNA with a gap size of about 7 bases. We suggest that the use of two DNAs has provided an unbiased view that RFC will most easily bind a gapped DNA with a minimal gap size of 5 bases, which will need an additional two 3' base pairs unwound to form an ideal 7-base gapped DNA substrate. Longer gaps could likely be easily accommodated as the ssDNA region could simply loop outside of RFC.

## Biochemical analysis of RFC−PCNA binding different sized gaps

To examine the gapped DNA binding of RFC−PCNA, we first measured binding affinities of RFC alone to gel-purified DNAs with various gap sizes in the presence of ATPγS. One may anticipate that the smallest gap that gives the tightest affinity may reflect the optimal gap size. Gapped DNAs that have a smaller ssDNA gap (i.e. <5 nucleotides) than needed to fit within the DNA sites of RFC require some DNA melting to increase their ssDNA gap size. The binding energy likely drives the 'melting' of the edges of small gaps to form a 5–6 nucleotide ssDNA gap. This expenditure of binding energy to melt DNA implies that the observed affinity to suboptimal gap sized DNAs is weaker than the actual value to an optimal gap size.

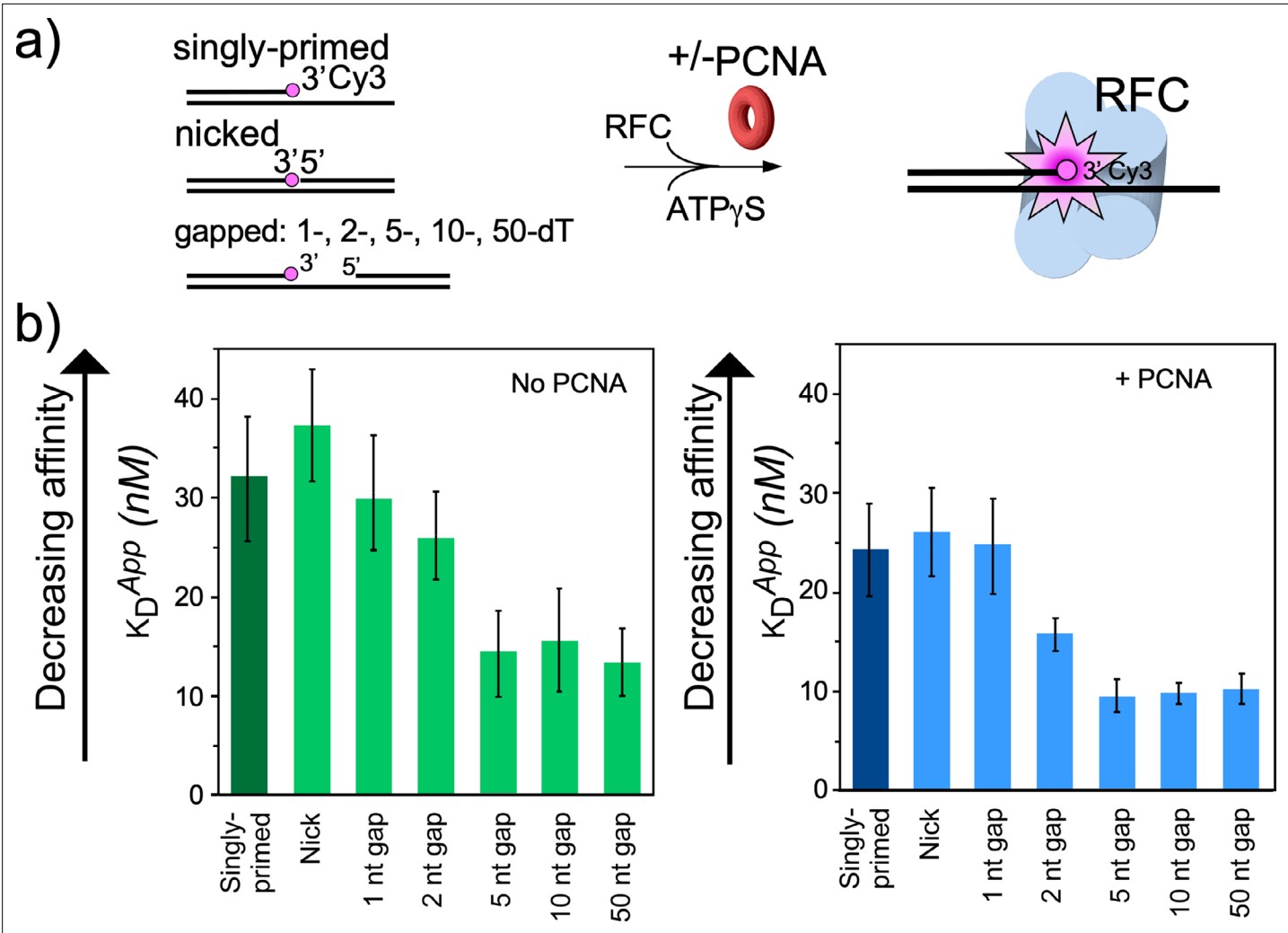

**Figure 4.** RFC/PCNA affinity for DNAs with different gap sizes. (**a**) Scheme for the affinity assay of RFC +/-PCNA for gapped DNAs. The top strand is composed of two constant length DNA oligonucleotides, that are hybridized to a variable length bottom strand to generate different sized gaps. The left most oligonucleotide had a 3' fluorophore. (**b**) Histograms for $K_D$ measurements of RFC binding assays to each DNA template +/-PCNA in the presence of ATPγS. The histograms in the absence of PCNA are shown to the left, and in the presence of PCNA are shown to the right. The raw titration data are shown in *Figure 4—figure supplements 3 and 4*. The data are obtained from analysis of independent triplicate assays, and the error bars show the standard deviation. The arrows indicate that the larger the $K_D$ value, the weaker the binding.

The online version of this article includes the following figure supplement(s) for figure 4:

**Figure supplement 1.** SDS-PAGE of proteins used in this report.

**Figure supplement 2.** Sequence of DNA oligonucleotides used for in vitro DNA binding assays using fluorescence.

**Figure supplement 3.** Binding of RFC to gapped DNA.

**Figure supplement 4.** Binding of RFC/PCNA to gapped DNA.

**Figure supplement 5.** Binding of RFC to 10-nt gap DNA in the absence of nucleotide compared to use of ATPγS.

**Figure supplement 6.** A 5' P at a 10-nt gap has no effect on RFC or RFC–PCNA binding.

**Figure supplement 7.** RFC binds a gapped DNA tighter than a mutant RFC lacking the 5'-DNA domain in Rfc1.

For binding experiments, we used a 3' oligo having a fluorophore, as in our earlier studies (*Johnson et al., 2006*). The proteins and the DNA oligonucleotides used for this work, are shown in *Figure 4— figure supplements 1–2*, respectively. The fluorescence intensity change during titration of RFC into the DNA was measured to determine the apparent $K_D$ of RFC binding to primed DNA, as well as DNAs containing either a nick or gaps of 1, 2, 5, 10, or 50 nt. We found that RFC binds all of the DNAs but exhibits an approximately two-fold tighter affinity (i.e. lower apparent $K_D$) for DNAs having

a gap of five bases or greater compared to singly primed DNA and DNA with a nick or 1 or 2 base gaps (*Figure 4a*, *Figure 4—figure supplement 3*). The cryo-EM structure predicted an optimal gap of 7 bases, but when one takes into account that RFC melts 2 bp at the 3′ end, the optimal binding to a 5-nucleotide gap is consistent with the cryo-EM data. RFC binds gaps larger than five nucleotides with comparable affinity, likely due to simply looping out the extra single-strand DNA. The same binding experiments in the presence of PCNA showed a similar outcome, except the affinities were all tighter (lower apparent $K_D$), likely due to the assistance of PCNA that binds RFC and holds it to DNA (*Figure 4b*, *Figure 4—figure supplement 4*). As a control, we use a 10-nt gapped DNA to confirm that nucleotide is required for RFC to bind tightly to DNA (*Figure 4—figure supplement 5*).

The gapped DNAs used here contained a 5′-OH. The RFC–PCNA–DNA structure reveals a region (formed by Rfc1 Lys-208, His-556 and His-659, *Figure 3d* and *Figure 3—figure supplement 1a, b*) in the 5′-DNA2 site having positively charged side chains that might bind the 5′-P of a 5′ ss/ds duplex DNA. Indeed, early studies indicated human and *Drosophila* BRCT regions bind recessed 5′-P DNA tighter than recessed 5′-OH DNA (*Allen et al., 1998*; *Kobayashi et al., 2006*). Therefore, we compared the binding affinity of RFC to a 10-nt gap in which the 5′ oligo used to form that gap has either a 5′-OH or 5′-P (*Figure 4—figure supplement 6*). The apparent $K_D$ values were not measurably different for a 5′-P versus a 5′-OH. On hindsight, this result seems reasonable considering that the 5′-DNA2 site on Rfc1 binds 10 bp of dsDNA with numerous contacts to Rfc1 (described above), that would overshadow the affinity derived from a single 5′-P.

To gain additional evidence that the difference in apparent $K_D$ values for gaps versus nicked or singly primed DNA was due to RFC binding a 5′ ss/dsDNA duplex, we compared the affinity of full-length RFC with RFC$^{\Delta N282}$, a truncated RFC (tRFC) that contains an Rfc1 lacking the amino terminal 282 amino acids including the BRCT domain. As expected, the RFC$^{\Delta 282}$ bound a 10-nt gapped DNA with similar affinity to a singly primed DNA compared to the approximately two-fold enhanced affinity of wt RFC containing the Rfc1 5′-DNA site (*Figure 4—figure supplement 7*).

## PCNA loading analysis at a singly primed DNA versus a 10 or 30-nt gap DNA +/- RPA

We next compared PCNA loading at a singly primed site lacking a downstream 5′ P-duplex compared to the presence of a downstream 5′ P-duplex that provides either a 10-nt or 30-nt ssDNA gap. To monitor PCNA loading, we labeled PCNA with $^{32}$P via an N-terminal six-residue kinase motif as we have described in detail earlier (*Kelman et al., 1995*). Since PCNA slides off linear DNA after it is loaded, we blocked PCNA sliding off DNA by attaching the biotinylated end to magnetic streptavidin beads, and the other end of the template strand contained a digitoxin (DIG) moiety, to which we added an antibody to DIG (see scheme in *Figure 5a*). The DNA sequences used for this work are in *Figure 5—figure supplement 1*. We then added $^{32}$P-PCNA and ATP (+/-RPA) and enabled loading for 30 s before quenching with EDTA. Magnetic beads were then washed and the retained $^{32}$P-PCNA was removed with SDS and analyzed by liquid scintillation. The overall results show that the downstream 5′-P-duplex (i.e. primer #2) facilitates loading of PCNA at both 10-nt and 30-nt gaps whether RPA is present or not (*Figure 5b*). This result is consistent with the 5′-DNA2 site within RFC facilitating PCNA loading at a short ssDNA gap. We note that RPA facilitates PCNA loading at a singly primed site, consistent with early results (*Tsurimoto and Stillman, 1991*), and this might be assisted by a direct interaction between RPA and RFC (*Yuzhakov et al., 1999*). The 10-nt gap is not noticeably stimulated by RPA, possibly due to the different mode of RPA interaction with a very small gap (*Figure 5b*). However, a 30-nt gap is optimal for RPA binding (*Chen and Wold, 2014*) and RPA appears to stimulate PCNA loading at the 30-nt gap (*Figure 5b*).

## Discussion
### Function of the 5′-DNA2 site of RFC in DNA repair

The fact that 3′-DNA1 and 5′-DNA2 are arranged in RFC like a single gapped DNA, as revealed by our structures, indicates a DNA gap repair function of the 5′-DNA2 binding site. We suggest that the 5′-DNA2 binding site functions in concert with the 3′-DNA1 site in nucleotide excision repair. DNA damage caused by base methylation or bulky UV-induced pyrimidine dimers and chemical carcinogen-induced adducts (*Cadet et al., 2005*; *Nohmi et al., 2005*) trigger the base and nucleotide

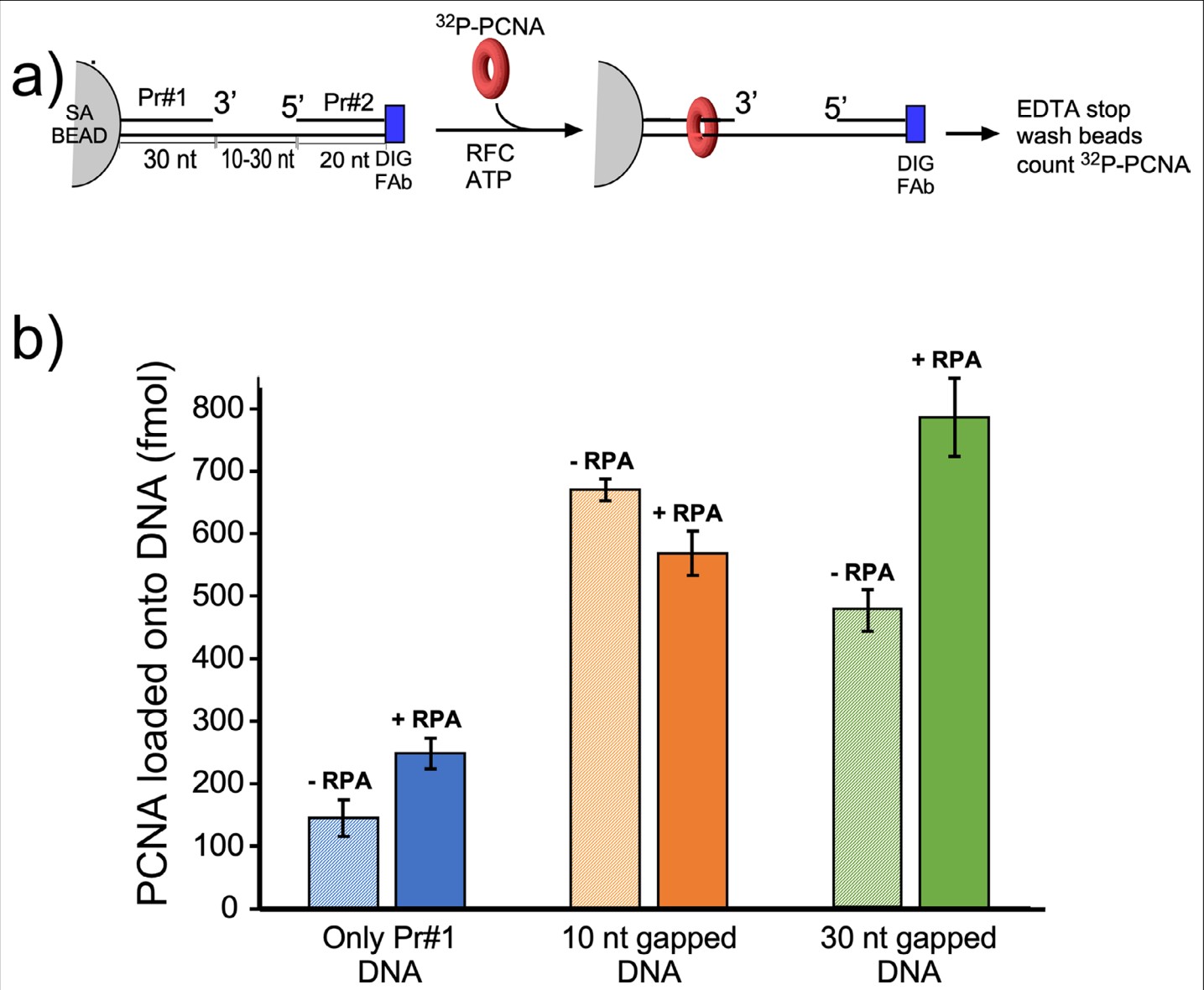

**Figure 5.** Efficiency of RFC loading PCNA onto gapped DNAs versus singly primed DNA. (**a**) Scheme of the PCNA loading assay. The DNA template strand contains a 3′ biotin and a 5′ Dig moiety. The DNA's were bound to streptavidin magnetic beads, then incubated with antibody to DIG to prevent PCNA sliding off DNA. Then RFC and $^{32}$P-PCNA were added with ATP for 30 s, quenched with EDTA, and beads were washed two times and $^{32}$P-PCNA that had been loaded onto DNA was counted by liquid scintillation. (**b**) Amounts of $^{32}$P-PCNA loaded by RFC onto either a singly primed DNA, a 10-nt gap DNA, or a 30-nt gap DNA in the presence or absence of RPA. All experiments were performed by independent triplicate assays and the error bars represent the SEM.

The online version of this article includes the following figure supplement(s) for figure 5:

**Figure supplement 1.** Sequence of DNA oligonucleotides used to load DNA onto streptavidin magnetic beads for in vitro PCNA loading.

excision repair pathways to remove lesion-containing bases, leaving a gap in DNA (**Boiteux and Jinks-Robertson, 2013**). Pol δ, Pol ε, Pol β, or possibly translesion (TLS) Pols are recruited by PCNA to fill-in the gap (**Overmeer et al., 2010**). Furthermore, early genetic studies demonstrated that mutants in the N-terminal region of Rfc1 are defective in excision gap repair (**Aboussekhra et al., 1995**; **Gomes et al., 2000**; **Li et al., 1994**; **McAlear et al., 1996**; **Shivji et al., 1992**). These genetic results are supported by our demonstration that RFC binds and loads PCNA with higher affinity and rate at gapped DNA than a single 3′-recessed primed template DNA. Our proposal is consistent with the

several previous reports indicating that the N-terminal region of Rfc1 is important for DNA damage gap repair, as further explained below.

## Possible coordination between the 3' and 5' sites in RFC repair activity

Previous genetic and structural studies have provided evidence that the N-terminal region of Rfc1 (i.e. the N-terminal DNA binding BRCT domain/ligase homology region) binds 5′ DNA and is demonstrated by genetic experiments to be important to DNA repair (*Aboussekhra et al., 1995*; *Allen et al., 1998*; *Gomes et al., 2000*; *Li et al., 1994*; *Shivji et al., 1992*; *Uchiumi et al., 1999*), for which PCNA is required (*Aboussekhra et al., 1995*; *Li et al., 1994*; *Shivji et al., 1992*). The predominant evidence for involvement of the N-terminal Rfc1 in repair are studies of methylated nucleotide bases using methylmethane sulfonate (MMS), that mostly methylates G, forming 7-methylguanine (7MeG), but also methylates A, forming 3-methyladenine (3MeA) (*McAlear et al., 1996*). These repair defects are well documented in a double mutant of Rfc1 in yeast at amino acids G185E and P234L (*McAlear et al., 1996*). In support of these findings, a Rfc1 mutant containing an insertion in the 5′ DNA binding region of Rfc1 also showed increased sensitivity to DNA damaging reagents (*Xie et al., 1999*). It is demonstrated that alkylated bases are repaired by base excision repair (*Xiao and Samson, 1993*). PCNA (and therefore also RFC) is required for base excision repair extending over 1 bp (*Aboussekhra et al., 1995*; *Li et al., 1994*; *Shivji et al., 1992*). It is proposed that the involvement of PCNA is not for processivity, given the short gap size of this type of repair, but is instead utilized to attract DNA polymerases (*Aboussekhra et al., 1995*; *Li et al., 1994*; *Shivji et al., 1992*). While DNA polymerases δ, and maybe ε appear to be involved in gap repair (*Blank et al., 1994*; *Wang et al., 1993*), the process is particularly mutagenic suggesting participation of DNA polymerase(s) with lower fidelity such as Pol β (yeast Pol IV) or error-prone translesion DNA Pols (e.g. Pol ζ -Rev1 and Pol η ) which can function with PCNA (*McAlear et al., 1996*). Another report on the physiological effect of deleting the N-terminal region of Rfc1 that binds 5′ dsDNA showed 2–10 fold reduction in cell growth upon MMS induced damage (*Gomes et al., 2000*; *McAlear et al., 1996*; *Xie et al., 1999*), but these results are smaller compared to earlier studies showing greater differences in Rfc1 mutants (*Gomes et al., 2000*;

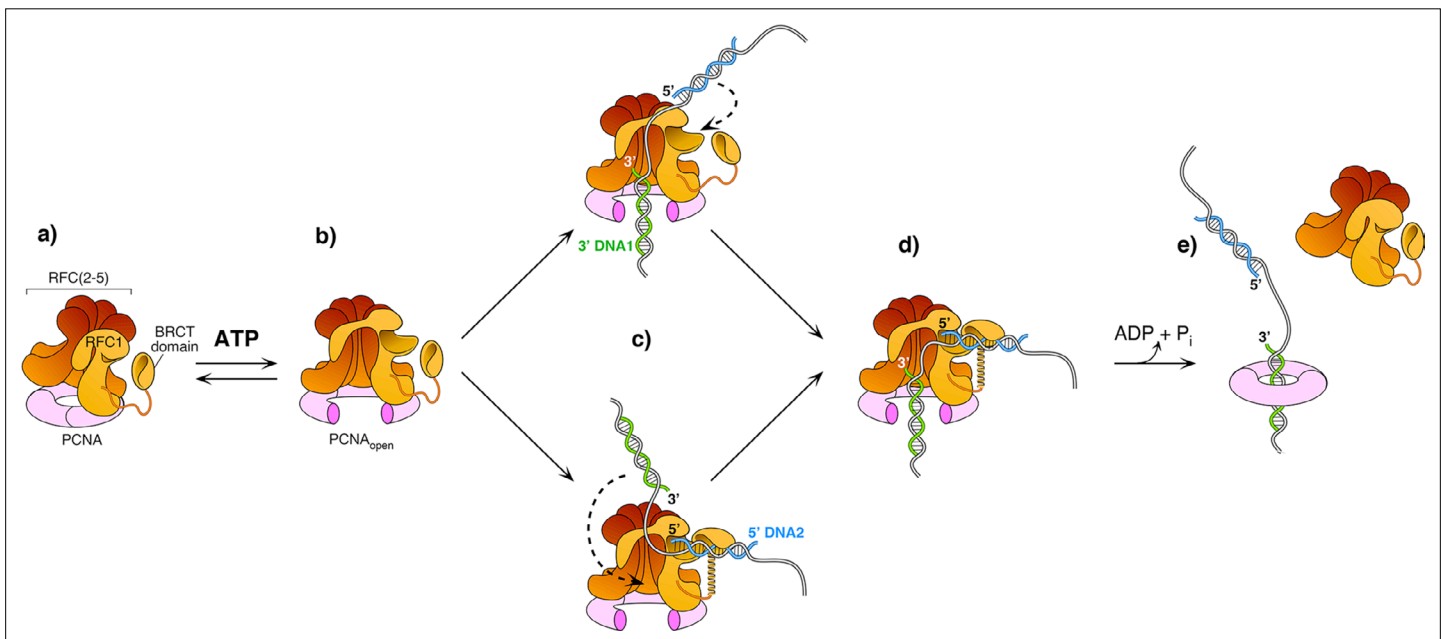

**Figure 6.** Model of RFC using both 3' and 5' DNA sites to load PCNA at a gap.
 (**a–b**) The ATPγS bound RFC opens PCNA and causes a conformation change that opens the A gate and PCNA, as well as exposing the shoulder 5'-DNA2 site (*Gaubitz et al., 2022*). (**c**) Once the A gate (for 3'-DNA1) and shoulder site (for 5'-DNA2) are exposed, either of two paths may be taken for RFC–PCNA binding to the 3' and 5' ends of a ssDNA gap. The top path shows 3'-DNA binding first, followed by 5'-DNA binding. The bottom path shows 5'-DNA binding followed by 3'-DNA binding. Regardless of which path is taken, after DNA binds in one site, filling of the remaining site is intramolecular and thus is probably very rapid. (**d-e**) Upon fully binding to the gapped DNA, RFC hydrolyzes ATP and dissociates from the clamp, enabling downstream enzymes, such as DNA Pols (not illustrated), to bind the PCNA encircling the gapped DNA.

*McAlear et al., 1996*; *Xie et al., 1999*). Perhaps, the difference lays in the use of single copy Rfc1 gene mutations in the chromosome in the earlier studies versus mutant RFC genes on a multiple copy centromeric plasmid. Nevertheless, both studies show that the 5′ DNA binding of RFC has a measurable effect on base excision repair of short gaps.

## Mechanism of RFC in loading PCNA onto a gapped DNA

The current study captured two DNA molecules bound to the two distinct 3′ and 5′ sites in RFC. The position and orientation of the two DNAs strongly implied RFC functions at a gap, and the biochemical studies confirm that a 7-nt gap is an optimal substrate for RFC. Taken together with a recent report while this study was under review (*Gaubitz et al., 2022*), we can propose a process as illustrated in *Figure 6*. Biochemical studies have shown that ATP binding enables clamp loader-clamp binding and opening in both eukaryotes and bacterial systems, illustrated in the first two diagrams of *Figure 6* (*Hingorani and O'Donnell, 1998*; *Kelch et al., 2012b*; *Kelch et al., 2012b*; *Turner et al., 1999*; *Zhuang et al., 2006*). Recent structural studies of RFC−PCNA−ATPγS reveal that the RFC−PCNA-open form is sufficiently wide to admit dsDNA into the 3′-DNA1 site (top path in *Figure 6c*; *Gaubitz et al., 2022*). The RFC−PCNA-open form structure also shows a competent shoulder 5′-DNA2 site, which could enable DNA to bind the shoulder 5′-DNA2 before DNA binds the 3′-DNA1 site (bottom path in *Figure 6c*; *Gaubitz et al., 2022*). While we observed binding of 3′-DNA1 before 5′-DNA2, this may be due to use of a linear DNA that could enter the 3′-DNA1 site either from the bottom (N-terminal) surface of RFC, or through the side of RFC via the A-gap. In the cell, primed 3′ DNA can only enter RFC via the side A-gap. Thus, analysis of DNA binding site order may require use of longer, or otherwise modified DNAs that cannot enter RFC through the 'bottom' N-face and must enter through the A gate.

Regardless of DNA site binding order, once one DNA site is bound, DNA binding to the other DNA site is an intramolecular reaction and thus probably quite rapid. Hence, the actual order of DNA site occupancy may be difficult to determine with precision. Structural work of this study and others (*Gaubitz et al., 2022*) show that the clamp can close around dsDNA without ATP hydrolysis, while ensemble studies have indicated that stable clamp closure occurs after ATP hydrolysis (*Anderson et al., 2009*; *Bowman et al., 2004*; *Liu et al., 2017*; *Marzahn et al., 2015*; *Sakato et al., 2012b*; *Trakselis et al., 2003*). Both the structural and bulk results may be correct considering that stable closure of the clamp may require ATP dependent RFC dissociation. Furthermore, several studies in bacterial and eukaryotic systems demonstrate that ATP hydrolysis is triggered at the end of the reaction, causing release of the clamp loader (last step in *Figure 6 Anderson et al., 2009*; *Bowman et al., 2004*; *Liu et al., 2017*; *Marzahn et al., 2015*; *Sakato et al., 2012b*; *Trakselis et al., 2003*).

We note that early studies using the isolated BRCT domain of RFC1 indicate a very tight affinity ($K_d$ of 10 nM) to recessed 5′ phosphate DNA (*Kobayashi et al., 2010*; *Kobayashi et al., 2006*). Combined with the very different structure predicted for the BRCT domain from NMR studies (*Kobayashi et al., 2010*; *Kobayashi et al., 2006*), it is possible that the free BRCT domain has a unique structure that binds 5′-P DNA tightly, and then changes its conformation upon binding to the shoulder of Rfc1. Consistent with this speculation, we see very little difference (i.e. about two-fold) in binding to DNA of RFC and RFC lacking the N-terminal BRCT region. Indeed, a conformation change that favors a lower affinity of RFC to DNA may facilitate the ejection of RFC from PCNA−DNA that is needed for PCNA to bind other factors.

## Is the 5′-DNA2 site a general feature of all DNA clamp loaders?

Given that both RFC and Rad24-RFC have a shoulder 5′ DNA binding site located in subunit A (i.e. Rfc1 or Rad24), the subunit that binds 5′ DNA in their respective clamp loaders, is it possible that other alternative clamp loaders, such as. Ctf18-RFC and Elg1-RFC (*Bellaoui et al., 2003*; *Naiki et al., 2001*; *Stokes et al., 2020*; *Yao and O'Donnell, 2012*), as well as the bacterial β-clamp loader or the T4 phage clamp loader, also possess the shoulder 5′ DNA binding site? Both Ctf18 and Elg1, in the subunit A position of RFC, likely contain an intact AAA+ module and a similar collar domain that resemble Rfc1 and Rad24, based on AlphaFold structure prediction (*Figure 2—figure supplement 2e, f*). The subunit in the analogous position to Rfc1 and Rad24 in the bacterial β-clamp loader is the δ subunit. The δ subunit maintains an intact AAA+ module and collar domain (*Simonetta et al., 2009*), and may contain a 5′ DNA binding site. Therefore, the eukaryotic alternative loaders Ctf18-RFC and

Elg1 likely possess the shoulder 5′ DNA binding site, and the bacterial loader might contain a 5′ DNA site. However, the subunit A (gp62) of the T4 phage clamp loader has lost much of the typical AAA+ module, retaining only three α-helices (*Kelch et al., 2011*), and is less likely to contain the shoulder DNA binding site (*Figure 1—figure supplement 4c*). Further biochemical and structural studies are needed to answer this question.

## Ideas and speculation

### Does the 5′-DNA2 site of RFC also function in lagging strand DNA replication?

An Okazaki fragment is about 150–200 nt long (*Ohashi and Tsurimoto, 2017*). Our biochemical data indicate RFC clamp loading activity is stimulated by a 5′ end at a 30-nt gap in the presence of RPA. Therefore, binding to both 3′ and 5′ DNA sites is not confined to an optimal 5–7 nt gap size determined by structural analysis alone. This opens the possibility that RFC maintains some level of contact with the 5′ end at gaps of larger sizes that can be bound by RPA, such as 150–200 bp Okazaki fragment gaps. One may speculate that an RFC–5′-DNA interaction could protect and prevent premature 5′ exonuclease excision at the 5′ end of Okazaki fragments. The two-site interaction of RFC, with the 3′ and 5′ ends of DNA, might also aid RFC processivity on the lagging strand in which RFC might transfer among multiple Okazaki fragments by walking from one to another during replication. It is also possible that there is some function of RFC in 5′-DNA end binding in DNA replication and/or DNA repair that is currently unknown. These possibilities will be a subject of future studies.

### Might RFC regulate the 9-1-1 response?

The reported cellular number of Rad24 is five-fold less than that of Rfc1, in which the shared Rfc2-5 subunits are far more abundant according to the proteomics in the S.c. database (https://www.yeast-genome.org as of Jan 20, 2022). It seems possible that RFC/PCNA represses the ability of Rad24-RFC to function in the 5′ loading of the 9-1-1 clamp at Okazaki fragments, and even at DNA repair gaps. In this speculation, and given the reported 5-fold higher concentration of Rfc1 over Rad24 protein in the cell, the ATR checkpoint kinase may not be stimulated by the 9-1-1 clamp until significant DNA damage has occurred, overwhelming RFC protection of 5′ ends and PCNA loading for DNA damage gap fill-in. Specifically, at low DNA damage, RFC may load PCNA for gap fill in, but when DNA damage is severe, the RFC might be insufficient to act at all the gaps - at which time the Rad24-RFC may have access to recessed 5′ ends and load 9-1-1 to turn on the DNA damage checkpoint. Thus, while quite speculative, RFC might act as a buffer against the loading of 9-1-1 for checkpoint activation. At some critical point, RFC will be overwhelmed by the number of repair gaps, and then Rad24-RFC will load the 9-1-1 clamp to upregulate the ATR kinase and induce a cell cycle checkpoint, enabling DNA repair until the genome is safe to continue being duplicated.

### What role does the unique subunit in alternative clamp loaders play?

Eukaryotes have four different clamp loaders, RFC, Rad24-RFC, Ctf18-RFC and Elg1-RFC, each of which contain Rfc2-5 subunits, but utilize a different large subunit that associates with RFC2-5 (*Lee and Park, 2020*). We and others have solved structures of Rad24-RFC-911 clamp-DNA (*Castaneda et al., 2022*; *Zheng et al., 2022*) revealing a 5′ DNA 'shoulder site' in Rad24, and show here that RFC can also bind 5′ DNA at a shoulder site in Rfc1. However, RFC primarily targets 3′ ss/ds DNA junctions to load the PCNA clamp for DNA polymerases and other proteins *Kelch et al., 2012b*; while Rad24-RFC primarily targets a 5′ ss/ds DNA junction for loading the 9-1-1 clamp to stimulate the ATR kinase (*Ellison and Stillman, 2003*; *Majka et al., 2006*; *Majka and Burgers, 2003*). Given these distinct DNA structural elements that are bound by the different clamp loaders, it is proposed that the other alternative clamp loaders may recognize different DNA structures for function to load or unload PCNA from DNA (*Bylund and Burgers, 2005*; *Fujisawa et al., 2017*; *Kubota et al., 2015*). However, further studies are required to more deeply understand the cellular functions of these four distinct eukaryotic clamp loaders. The structure and individual functions of the four different eukaryotic 'clamp loaders' is a subject that remains a fascinating avenue for future research.

## Materials and methods

### Reagents and proteins

Radioactive nucleotides were from PerkinElmer Life Sciences (Waltham, Massachusetts). Unlabeled ATP was from Cytiva (Marlborough, MA). ATPγS was from Roche (Basel, Switzerland). DNA-modification enzymes were from New England BioLabs (Ipswich.Massachusetts). DNA oligonucleotides were from Integrated DNA Technologies (Coralville, Iowa). Protein concentrations were determined with the Bio-Rad Labs (Hercules, California) Bradford Protein stain using bovine serum albumin as a standard. Streptavidin-coated Dynabead M-280 magnetic beads were purchased from Thermo-Fisher Scientific (Waltham, Massachusetts). Anti-Digoxigenin, the anti-Dig Fab, was from Millipore-Sigma (St. Louis, MO).

### Cell growth

Expression plasmids were transformed into BLR(DE3) (Novagen, Madison, Wisconsin) *E. coli* - competent cells and selected on LB plates containing ampicillin (100 µg/ml). When needed for use of two expression plasmids, two plasmid transformants required selection on both Amp (100 µg/ml) and kanamycin (50 µg/ml). Fresh transformants were grown in 12–24 L of LB media containing the appropriate antibiotics at 30 °C until reaching an $OD_{600}$ value of 0.6. The cultures were then quickly brought to 15 °C by swirling culture flasks in an ice water bath with a thermometer, then placed into a prechilled shaker incubator at 15 °C before adding 1 mM isopropyl-1-thio-β-d-galactopyranoside. Induced cells were then incubated at 15 °C for ~18 hr. Cells were collected by centrifugation and resuspended in an equal weight of Tris-Sucrose (50 mM Tris-HCl pH7.5, 10% sucrose w/v) and frozen until use. Frozen cells were thawed and then lysed by two passages through an Emulsi Flex-C-50 from Avestin (Ottawa, ON, Canada) using pulses of 20,000 psi and 5000 psi at 4 °C. Cell debris was removed by centrifugation (19,000 r.p.m. in a SS-34 rotor for 1 h at 4 °C).

### PCNA

The expression plasmid for PCNA was generated by PCR of the *pol30* gene from yeast genomic DNA and then cloned into a modified pET16 vector containing a hexahistine tag and 6 residue site for the catalytic subunit of cAMP dependent protein kinase A at the N-terminus of the expressed protein as described (*Kelman et al., 1995*). Forty-eight L of transformed cells were grown as described above and resuspended in Buffer I (20 mM Tris-HCl (pH 7.9), 500 mM NaCl) plus 5 mM imidazole. The following procedures were performed at 4 °C. After cell lysis and clarification by centrifugation, the supernatant (8 g) was loaded onto a 150 ml HiTrap Ni chelating resin (Cytiva, Marlborough, MA) and washed with 300 ml Buffer I containing 60 mM imidazole. Following this, protein was eluted using a 1.6 L linear gradient of Buffer I from 60 mM to 500 mM imidazole, collecting 25 ml fractions. Presence of proteins was followed in 10% SDS-polyacrylamide gels stained with Coomassie Blue. Fractions 10–38 were pooled (3.4 g) and dialyzed against Buffer A (20 mM Tris-HCl, pH 7.5, 5 mM DTT, 0.1 mM EDTA and 10% glycerol). The dialysate was loaded onto a 200 ml DEAE Sepharose column equilibrated in Buffer A. The column was washed with 500 ml Buffer A, then protein was eluted using a 2 L linear gradient of Buffer A from zero NaCl to 500 mM NaCl; fractions of 22 ml were collected. Presence of proteins was followed in 10% SDS-polyacrylamide gels stained with Coomassie Blue. Fractions 54–70 were pooled to give 1.8 g>95% pure PCNA. The pool was dialyzed against Buffer A, and a small amount of precipitate was removed by centrifugation, giving 1.4 gram of PCNA which was then aliquoted, snap frozen in liquid $N_2$, and stored at –80 °C.

### RFC

Full-length *S. cerevisiae* RFC utilized two expression plasmids having different and compatible origins as described previously (*Finkelstein et al., 2003*). pLANT-2/RIL–RFC[1+5] was co-transformed with pET(11 a)-RFC[2+3 + 4] into BLR(DE3) cells (Novagen, Madison, Wisconsin). The cell lysate was clarified by centrifugation, diluted with Buffer A to ~150 mM NaCl, and then applied to a 30 ml SP-Sepharose Fast Flow column (Cytiva, Marlborough, MA) equilibrated with Buffer A containing 150 mM NaCl. The column was eluted with a 300 ml gradient of 150–600 mM NaCl in Buffer A. Presence of proteins was followed in 10% SDS-polyacrylamide gels stained with Coomassie Blue. The peak of RFC (which eluted at ~365 mm NaCl) was pooled and diluted with Buffer A to ~150 mM NaCl. The

protein was then applied to a 40 ml Q-Sepharose Fast Flow column (Cytiva, Marlborough, MA) equilibrated with Buffer A containing 150 mM NaCl and eluted with a 400 ml gradient of 150–600 mM NaCl in Buffer A. The fractions containing RFC complex (which eluted at ~300 mM NaCl) were pooled (24 mg), aliquoted, flash frozen in liquid $N_2$ and stored at –80 °C. Truncated RFC$^{\Delta N282}$ was purified by the same procedure.

### RPA: RPA was expressed and purified as described

An SDS-PAGE analysis of these protein preparations is shown in *Figure 4—figure supplement 1*, (*Henricksen et al., 1994*).

## Cryo-EM grids preparation and data collection

The double-tailed DNA substrate with both 3'- and 5'-recessed ends was composed of a 20-nt primer strand (5'- GCA GAC ACT ACG AGT ACA TA –3') and a 40-nt template strand (5'- TTT TTT TTT TTA TGT ACT CGT AGT GTC TGC TTT TTT TTT T –3'). They were synthesized, HPLC purified, and then annealed by Integrated DNA Technologies Inc. The DNA substrate is also referred to as P/T_DNA in the manuscript. The in vitro assembly of yeast RFC–PCNA–DNA complex followed our previous procedure for assembling the S.c. Pol δ–PCNA–DNA complex (*Zheng et al., 2020*). Briefly, 1 μl of purified PCNA clamp at 30 μM, 3.0 μl of double tailed DNA at 100 μM were mixed and incubated at 30 °C for 10 min, then the mixture along with 0.75 μl 10 mM ATPγS and 0.75 μl 100 mM Mg-Acetate were added into 10.0 μl of purified RFC protein at 3.3 μM concentration for a final concentration of 2.2 μM RFC, 2.0 μM PCNA clamp, 20.0 μM DNA, 0.5 mM ATPγS, and 5 mM Mg-Acetate, in a total reaction volume of 15 μl. The final molar ratio of RFC: PCNA clamp: DNA was 1.0: 0.9: 9.0. The mixture was then incubated in an ice-water bath for an additional 1 hr. The Quantifoil Cu R2/1 300 mesh grids were glow discharged for 1 min in a Gatan Solarus, then 3 μl of the mixture was applied onto the EM grids. Sample vitrification was carried out in a Vitrobot (Thermo Fisher Mark IV) with the following settings: blot time 2 s, blot force 4, wait time 2 s, inner chamber temperature 6 °C, and a 95% relative humidity. The EM grids were flash-frozen in liquid ethane cooled by liquid nitrogen. Cryo-EM data were automatically collected on a 300 kV Titian Krios electron microscope controlled by SerialEM in a multi-hole mode. The micrographs were captured at a scope magnification of 105,000×, with the objective lens under-focus values ranging from 1.3 to 1.9 μm, by a K3 direct electron detector (Gatan) operated in the super-resolution video mode. During a 1.5 s exposure time, a total of 75 frames were recorded with a total dose of 66 e$^-$/Å$^2$. The calibrated physical pixel size was 0.828 Å for all digital micrographs.

## Image processing and 3D reconstruction

The data collection and image quality were monitored by the cryoSPARC Live v3.2 (*Punjani et al., 2017*) installed in a local workstation. The image preprocessing including patch motion correction, contrast transfer function (CTF) estimation and correction, blob particle picking (70–150 Å diameter) and extraction with a binning factor of 2 were also achieved at the same time, and a total number of 16,401 raw micrographs were recorded during a 3-day data collecting session. We performed two rounds of 2-dimensional (2D) image classification, which resulted in ~1.3 million 'good' particle images. Then we trained Topaz (*Bepler et al., 2019*) and used the trained model to pick more particles. We next used the reported 'Build and Retrieve' method to avoid missing those less frequently occurring particle views (*Su et al., 2021*). The cleaned-up particles were combined, and duplicate particles with 40% overlapping (52 Å) or larger than the particle diameter (~130 Å) were removed. The final dataset contained ~2.9 million particles that was used for 3D reconstruction. We first calculated four starting 3D models and obtained two major 3D classes. In the first 3D class, RFC is clearly bound to DNA but not to PCNA. In the second 3D class, RFC is seen bound to both DNA and PCNA. We subjected both classes to non-uniform 3D refinement and 3D viability analysis (3DVA), resulting in eight 3D subclasses for each of the two starting 3D classes.

Among the eight 3DVA-derived subclasses of the RFC–DNA complexes in the absence of PCNA, we found two subclasses bound to one DNA (RFC–DNA1) and two subclasses bound to two DNA molecules (RFC–3'-DNA1–5'-DNA2). The remaining four subclasses contained 3'-DNA1 with partially occupied 5'-DNA2, and they were discarded without further processing. Particles in the two subclasses bound to 3'-DNA1 only were combined into a dataset with 334,876 particles, and they were refined to a 3D map of 3.33 Å average resolution. Particles in the two subclasses bound to 3'-DNA1 and 5'-DNA2

were also combined to a dataset of 315,619 particles, and they were refined to a 3D map at 3.25 Å average resolution.

Among the eight 3DVA-derived subclasses of the RFC–DNA–PCNA complex, 1 subclass bound to 3'-DNA1 only and the bound PCNA was in an open spiral conformation. This subclass contained 118,384 particles and was further refined to a 3D map at 3.41 Å average resolution (RFC–3'-DNA1–open ring PCNA). Two subclasses bound to 3'-DNA1 only with the PCNA in a closed ring conformation. Particles in these two subclasses were combined to a dataset of 166,348 particles, and they were refined to a 3D map at 3.30 Å average resolution (RFC–3'-DNA1–closed ring PCNA). We found 4 subclasses contained 2 DNA molecules, in which, 3 subclasses were combined into a dataset of 432,904 particles and refined to a 3D map (RFC–3'-DNA1–5'-DNA2–PCNA) at 3.09 Å average resolution. The two remaining subclasses only had partial PCNA density, were discarded without further processing. In the 4 subclasses bound to PCNA and 3'-DNA1 and 5'-DNA2, two subclasses did not contain density, one had weak density, and one had strong density for the Rfc1 BRCT domain. We performed focused refinement on 85,932 particles in the subclass with strong density for Rfc1 BRCT. We used a 3D mask that included the Rfc1 NTD (BRCT and AAA+ module) and 5'-DNA2 region for particle subtraction and refined a local 3D map at 3.90 Å average resolution. This local 3D map was aligned to the 3.09 Å 3D map of the full complex and combined in UCSF Chimera (*Pettersen et al., 2004*) to obtain the final composite map of the RFC–3'-DNA1–5'-DNA2–PCNA complex.

## Model building, refinement, and validation

We used the crystal structure of the yeast RFC–PCNA complex (PDB entry 1SXJ) as the initial model for atomic model building of the above described five 3D EM maps. The EM map of the Rfc1 BRCT domain was at a low resolution of 3.90 Å, and there was no available high-resolution structure for the yeast domain. We referenced the crystal structure of the human RFC1 BRCT (PDB entry 2K6G) and the AlphaFold predicted yeast Rfc1 model (AF-P35251-F1) to build the atomic model (*Jumper et al., 2021*). The DNA from the T4 phage clamp–loader–DNA crystal structure (PDB entry 3U5Z) and the DNA from yeast Rad24-RFC–9-1-1 clamp–DNA complex structure (PDB entry 7SGZ) were adopted as the initial models for chamber (3'-DNA1) and shoulder DNA (5'-DNA2) building, respectively. We also used the de novo modeling program Map-to-Model wrapped in PHENIX (*Adams et al., 2010*) and the module 'Automated Nucleic Acid building' function integrated in COOT (*Emsley et al., 2010*) to help modeling the two DNA molecules. These initial models are fitted into the composite 3D map of RFC–3'-DNA1–5'-DNA2–PCNA and merged into one single coordinate file in the UCSF Chimera (*Pettersen et al., 2004*) to serve as the starting model for the entire complex. The starting model was refined iteratively between the real space computational refinement in PHENIX and the manual adjustment in COOT. The atomic model of the RFC–3'-DNA1–5'-DNA2–PCNA complex was finally refined to 3.1 Å and went through a comprehensive validation by the MolProbity program (*Chen et al., 2010*) embedded in PHENIX. This model then served as the initial model of the other four 3D maps. The model for each 3D map was subjected to iteratively real space refinement and manually adjustment as described above. The two PCNA subunits at the gate had weaker EM densities in the spirally open PCNA EM map, and they were modelled by rigid body docking with a PCNA monomer structure. Structure figures were prepared using ChimeraX (*Pettersen et al., 2021*) and organized in Adobe Illustrator (Adobe Inc, San Jose, CA).

## Fluorescent DNA binding assays

The oligonucleotides used to make the DNA substrates (*Figure 4—figure supplement 2* and *Figure 5—figure supplement 1*) were synthesized and PAGE or HPLC-purified by Integrated DNA Technologies (Coralville, Iowa). Gapped DNA templates used in the fluorescent binding assays were obtained by annealing the Primer1-Cy3, Primer 2 (or Primer2 5'-P-DNA) with their corresponding complementary DNA template oligos (Tmpl01_40, Tmpl02_41, etc.). The annealed templates were purified from a 10% native PAGE in 0.5 x TBE; gel bands were cut out from the gel, extracted in TE buffer and further concentrated and buffer exchanged into standard TE buffer (10 mM Tris-HCl pH 7.5 and 1 mM EDTA).

RFC binding to DNA was measured using each 3' Cy3-labeled gel purified DNA template (singly primed, nicked, 1-, 2-, 5-, 10-, or 50-dT gaps) at constant 10 nM DNA concentration. RFC complex was titrated from 0 to 240 nM into reactions of 50 µL final volume containing 20 mM Tris-HCl (pH

7.5), 100 mM NaCl, 8 mM Mg-Acetate, 4 mM DTT, 0.1 mM EDTA and 100 μM ATPγS. When present, the PCNA trimer was at a concentration ratio of 1:1 with RFC. The reaction mixtures were assembled in a 384-well plate format, then incubated at room temperature for 20 min. The plates were centrifuged for 20 s to eliminate any bubbles and to assure that the liquid mix is evenly distributed on the well-bottom; the fluorescence signal was recorded using the Synergy Neo2 plate reader fluorimeter (BioTek Instruments, Winooski, VT) by exciting the Cy3 fluorophore at 535 nm and read out of the integrated fluorescence signal between 560 and 580 nm. The fluorescence recording routine involved a reading every 3 min preceded by an elliptical motion plate stirring for 20 s. The recording was performed for 45 min at constant temperature of 23.8 °C, for a total 16 readings averaged for each concentration point. Because the averages of the first four recordings and the last four readings are same in this 45 min regimen, the reactions are expected to be at equilibrium conditions. Three independent experiments were performed for each titration. Relative intensity ($I/I_0$) was plotted versus RFC concentration. Apparent $K_d$ measurements were determined using the quadratic model of a single-site interaction that more accurately accounts for bound protein (equation 5 in *Jarmoskaite et al., 2020*) corrected with a nonspecific binding parameter, using the OriginPro software (OriginLab Corporation, Northampton, MA, USA).

### $^{32}$P-PCNA clamp loading on gapped DNAs

For magnetic beads assays the template strand comtained a 5'-Digitoxin and a 3'-Biotin. The Dig/Bio template and the primer #1 and, when present, primer #2, were mixed with their respective template strand (*Figure 5—figure supplement 1*) in a 1:2 ratio of template to each primer in 50 μl of 5 mM Tris-HCl, 150 mM NaCl, 15 mM sodium citrate (final pH 8.5), then was incubated in a 95 °C water bath that was allowed to cool to room temperature over a 30 min interval. Streptavidin-coated Dynabead M-280 magnetic beads were purchased from Thermo-Fisher Scientific (Waltham, MA) and anti-Digoxigenin, the anti-Dig Fab (Millipore-Sigma (Burlington, MA)). Each primed biotinylated DNA template (200 pmol) was incubated with 1 mg Dynabeads M-280 Streptavidin in 5 mM Tris-HCl (pH 7.5), 0.5 mM EDTA and 1 M NaCl. An average yield of 130–200 pmol DNA/mg of Dynabeads, was determined by releasing DNA from beads by 1% SDS, removing beads, then measuring the absorbance of the supernatant at 260 nm using a nanodrop spectrophotometer. To block non-specific protein binding the DNA-bead conjugates were pre-incubated with 5 mg/ml BSA in 10 mM Tris-HCl (pH 7.5), 1 mM EDTA, 150 mM NaCl at room temperature for 30 minutes and washed in the same buffer until no BSA remained in the supernatant as detected by Bradford reagent.

The kinase tag on PCNA was used to label it with $^{32}$P to a specific activity of 40 cpm/fmol with [γ-$^{32}$P] ATP using the recombinant catalytic subunit of cAMP-dependent protein kinase produced in *E. coli* (a gift from Dr. Susan Taylor, University of California at San Diego) as described earlier (*Kelman et al., 1995*). Before the start of the PCNA-loading assay, either no RPA, or a 2.5-fold molar excess RPA over DNA was added and the 5' Dig of the DNA-bead conjugate was blocked with anti-digoxigenin Fab fragment in a ratio of 1:2 respectively in clamp loading buffer (30 mM HEPES-NaOH (pH 7.5), 1 mM DTT and 1 mM CHAPS) for 10 minutes at room temperature (23 °C). The clamp loading assays were then performed at 23 °C. Each reaction contained 167 nM antibody-blocked DNA-bead conjugate, 376 nM $^{32}$P-PCNA in 30 μl of clamp loading buffer, 8 mM MgCl$_2$, 1 mM ATP and was initiated with 167 nM RFC for 30 seconds before being quenched with 37 mM EDTA. The beads were then collected in a magnetic separator and washed 2 X with 200 μl clamp loading buffer containing 100 mM NaCl. The DNA-bound $^{32}$P-PCNA was stripped from the beads with 0.5% SDS and 5 minutes of boiling, then counted by liquid scintillation. The singly-primed template (ie. with only Primer #1 annealed) had a 5' 30-nt extension of ssDNA that could bind RPA since this DNA substrate shared the same template as the 10-nt gap DNA.

## Acknowledgements

Cryo-EM micrographs were collected at the David Van Andel Advanced Cryo-Electron Microscopy Suite in Van Andel Institute. We thank G Zhao and X Meng for facilitating data collection. This work was supported by the US National Institutes of Health grants GM131754 (to HL) and GM115809 (to MEO), Van Andel Institute (to HL), and Howard Hughes Medical Institute (to MEO).

## Additional information

### Competing interests
The other authors declare that no competing interests exist.

### Funding

| Funder | Grant reference number | Author |
|---|---|---|
| National Institute of General Medical Sciences | GM131754 | Huilin Li |
| National Institute of General Medical Sciences | GM115809 | Michael E O'Donnell |

The funders had no role in study design, data collection and interpretation, or the decision to submit the work for publication.

### Author contributions
Fengwei Zheng, Data curation, Formal analysis, Investigation, Methodology, Resources, Validation, Writing - original draft, Writing - review and editing; Roxana Georgescu, Nina Y Yao, Data curation, Formal analysis, Investigation, Methodology, Resources, Validation; Huilin Li, Conceptualization, Formal analysis, Funding acquisition, Investigation, Methodology, Project administration, Resources, Supervision, Validation, Writing - original draft, Writing - review and editing; Michael E O'Donnell, Conceptualization, Funding acquisition, Investigation, Methodology, Project administration, Resources, Supervision, Validation, Writing - original draft, Writing - review and editing

### Author ORCIDs
Fengwei Zheng ORCID http://orcid.org/0000-0002-7139-4831
Roxana Georgescu ORCID http://orcid.org/0000-0002-1882-2358
Huilin Li ORCID http://orcid.org/0000-0001-8085-8928
Michael E O'Donnell ORCID http://orcid.org/0000-0001-9002-4214

### Decision letter and Author response
Decision letter https://doi.org/10.7554/eLife.77469.sa1
Author response https://doi.org/10.7554/eLife.77469.sa2

---

## Additional files

### Supplementary files
• Transparent reporting form

### Data availability
The 3D cryo-EM maps of *S. cerevisiae* RFC−DNA and RFC−PCNA−DNA complexes have been deposited in the Electron Microscopy Data Bank with accession codes EMD-25872 (RFC−PCNA−DNA1−DNA2), EMD-25873 (RFC−open PCNA−DNA1), EMD-25874 (RFC−closed PCNA−DNA1), EMD-25875 (RFC−DNA1−DNA2), and EMD-25876 (RFC−DNA1). The corresponding atomic models have been deposited in the Protein Data Bank with accession codes 7TFH, 7TFI, 7TFJ, 7TFK and 7TFL.

The following datasets were generated:

| Author(s) | Year | Dataset title | Dataset URL | Database and Identifier |
|---|---|---|---|---|
| Zheng F, Georgescu RE, Yao YY, Li H, O'Donnell ME | 2022 | 3D cryo-EM map of RFC−PCNA−DNA1−DNA2 | https://www.ebi.ac.uk/pdbe/emdb/EMD-25872 | Electron Microscopy Data Bank, EMD-25872 |

*Continued on next page*

*Continued*

| Author(s) | Year | Dataset title | Dataset URL | Database and Identifier |
|---|---|---|---|---|
| Zheng F, Georgescu RE, Yao YY, Li H, O'Donnell ME | 2022 | 3D cryo-EM map of RFC–open PCNA–DNA1 | https://www.ebi.ac.uk/pdbe/emdb/EMD-25873 | Electron Microscopy Data Bank, EMD-25873 |
| Zheng F, Georgescu RE, Yao YY, Li H, O'Donnell ME | 2022 | 3D cryo-EM map of RFC–closed PCNA–DNA1 2022 | https://www.ebi.ac.uk/pdbe/emdb/EMD-25874 | Electron Microscopy Data Bank, EMD-25874 |
| Zheng F, Georgescu RE, Yao YY, Li H, O'Donnell ME | 2022 | 3D cryo-EM map of RFC–DNA1–DNA2 2022 | https://www.ebi.ac.uk/pdbe/emdb/EMD-25875 | Electron Microscopy Data Bank, EMD-25875 |
| Zheng F, Georgescu RE, Yao YY, Li H, O'Donnell ME | 2022 | 3D cryo-EM map of RFC–DNA1 | https://www.ebi.ac.uk/pdbe/emdb/EMD-25876 | Electron Microscopy Data Bank, EMD-25876 |
| Zheng F, Georgescu RE, Yao YY, Li H, O'Donnell ME | 2022 | Atomic model of RFC–PCNA–DNA1–DNA2 2022 | https://rcsb.org/structure/7TFH | Protein Data Bank, 7TFH |
| Zheng F, Georgescu RE, Yao YY, Li H, O'Donnell ME | 2022 | Atomic model of RFC–open PCNA–DNA1 | https://rcsb.org/structure/7TFI | Protein Data Bank, 7TFI |
| Zheng F, Georgescu RE, Yao YY, Li H, O'Donnell ME | 2022 | Atomic model of RFC–closed PCNA–DNA1 | https://rcsb.org/structure/7TFJ | Protein Data Bank, 7TFJ |
| Zheng F, Georgescu RE, Yao YY, Li H, O'Donnell ME | 2022 | Atomic model of RFC–DNA1–DNA2 | https://rcsb.org/structure/7TFK | Protein Data Bank, 7TFK |
| Zheng F, Georgescu RE, Yao YY, Li H, O'Donnell ME | 2022 | Atomic model of RFC–DNA1 | https://rcsb.org/structure/7TFL | Protein Data Bank, 7TFL |

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
