## [Editor Report]

The role of Replication Factor C (RFC) in DNA replication and repair has been known for many years. RFC/PCNA binds to a double strand-single strand DNA junction with a 3'-recessed end, with the DNA passing through a central chamber in the five-subunit protein. The current paper reports structures of RFC/PCNA with two separate DNA molecules, one containing the well characterized 3'-recessed DNA and surprisingly, a second 5'-recessed DNA outside the central chamber. The paper is an important addition to understanding RFC function, particularly in DNA repair.

---

## [Decision Letter]

**Decision letter after peer review:**

Thank you for submitting your article "A novel 5′ DNA binding site in RFC facilitates PCNA loading for gap DNA repair" for consideration by *eLife*. Your article has been reviewed by 3 peer reviewers, including Bruce Stillman as Reviewing Editor and Reviewer #1, and the evaluation has been overseen by Volker Dötsch as the Senior Editor. The following individuals involved in the review of your submission have agreed to reveal their identity: Peter Burgers (Reviewer #2); David Jeruzalmi (Reviewer #3).

Essential revisions:

1. The studies in Figure 5 report the affinity of gapped DNA with either a 10 base gap or a 50 base gap, the latter with RPA loaded. Since RPA has two DNA biding modes, binding to an 8 base ssDNA and a 30 base ssDNA, the experiments in Figure 5 should be repeated: in panel (b) adding RPA to the 10 base gap DNA and in panel (c) performing a reaction with no added RPA. The reason is that RPA could stimulate the loading of PCNA onto the 10 base gap DNA and it could stimulate the binding of RFC/PCNA specifically to the 3'-recessed ssDNA/dsDNA junction, as shown previously (Tsurimoto and Stillman, 1991, J. Biol Chem. 266: 1950-1960).

2. The use of the word "novel" in the title is not appropriate since the amino-terminus of RFC1 has been shown to bind DNA previously. This paper only clarifies how it binds and provides function. Words such as novel should be reserved for truly unique findings.

3. The observation of the ring closed form of RFC-ATPgammaS-DNA-PCNA is unexpected because it shows that closure can occur before hydrolysis and the main function of hydrolysis is the release of RFC. However, kinetic studies (e.g. a 2012 Hingorani JMB paper) show that with ATP, hydrolysis precedes closure and this should be clarified in the paper.

4. Biochemical studies with the isolated BRCT domain show that it binds partial double-stranded DNA with a 5'-phosphorylated junction with much higher affinity than the same DNA lacking the 5'-phosphate. The 1998 NAR Hardy study and the 2010 Siegal paper describing these studies have been referenced, but they were not discussed. Why were the cryoEM studies carried out with DNAs lacking the 5'-phosphate?

5. Can the orientation of the 5'-DNA molecule be defined as unambiguous in these structures? If so, and the 5'-OH is not engaged in interactions, what sets the orientation as 5' at the BRCT domain?

6. In Figure 3D, it appears that the 5'-junction does not contact specific residues in the BRCT. We need to see better views of these interactions. And the same for the structure with the open PCNA. This is the major point in the paper and it is not well presented. Interestingly, the model structure 2K7F from Siegal shows that the 5'-phosphate is engaged in binding positive residues, as perhaps could be expected from imposing a high binding energy for such an interaction in the modeling program. Could the authors comment on whether the chain direction for the second DNA molecule could be ascertained directly from the EM maps? Still, this entire, critical point deserves more attention than it has been given.

7. Two experiments have been included in supplementary 9 and 10, which indicate that the presence of phosphate does not affect DNA binding. But in these studies, the bulk of binding energy comes from the ATPgammaS-clamploader-PCNA interactions with DNA. What about the binding of RFC to DNA in the absence of ATP or ATPgammaS?

8. Biochemical analysis RFC-PCNA on DNA with nicks and varying length gaps.

a. The authors conclude that RFC binds tightly to all the substrates tested but binds with a two-fold (~15 nM vs ~30 nM) higher affinity for DNA molecules with gaps that have five or more bases.

b. The 5 nucleotide or large gaps compares well with the structural finding.

c. The presence of PCNA in the measurement reveals tighter binding than in its absence, but with the same trend as seen when PCNA was omitted.

d. Analysis of an RFC ensemble whose RFC1 lacks the BRCT domain revealed 2-fold tighter binding (16 nM vs. 32 nM) when BRCT was present than when not.

In view of the extensive set of contacts to DNA mediated by the BRCT domain, it is surprising that there is not a greater difference in Kd between primer-template and gapped DNA structures. Likewise, why does deletion of BRCT hardly change the affinity for DNA?

9. The Kd values for RFC (-/+ PCNA) binding of various DNA structures (Figure 4, etc) were calculated from experiments where RFC was titrated against a DNA concentration held fixed at 10 nM. Given that the measured Kds values were in the ~15-40 nM range, might the fixed concentration of DNA be too high to reveal an accurate Kd? Might this explain why little or difference in Kd was found?

If maintaining the fixed concentration of DNA at 10 nM is required by the experiment, other approaches to extracting (perhaps) more accurate Kds might be found here:

Jarmoskaite, I., AlSadhan, I., Vaidyanathan, P. P. and Herschlag, D. How to measure and evaluate binding affinities. *eLife* 9, e57264 (2020).

---

## [Author Response]

Essential revisions:1. The studies in Figure 5 report the affinity of gapped DNA with either a 10 base gap or a 50 base gap, the latter with RPA loaded. Since RPA has two DNA biding modes, binding to an 8 base ssDNA and a 30 base ssDNA, the experiments in Figure 5 should be repeated: in panel b adding RPA to the 10 base gap DNA and in panel c performing a reaction with no added RPA. The reason is that RPA could stimulate the loading of PCNA onto the 10 base gap DNA and it could stimulate the binding of RFC/PCNA specifically to the 3’-recessed ssDNA/dsDNA junction, as shown previously (Tsurimoto and Stillman, 1991, J. Biol Chem. 266: 1950-1960).

We appreciate the reviewer’s comments, and the new revised figure includes comparison of singly primed, and two gaps – a 10 nt gap and a 30 nt gap – all experiments are performed plus and minus RPA and in independent triplicate sets of assays. We also cite the Tsurimoto and Stillman citation given above. As the reviewer’s anticipated, RPA stimulates PCNA loading at a 3’ terminus, and also at a 30 nt gap to which RPA can bind in a normal mode. At a 10 nt gap, the RPA gave did not appear to stimulate PCNA loading. Overall, the results of the figure demonstrate the functionality of the 5’ recessed end in clamp loading for both sized gaps compared to singly primed DNA (i.e. lacking a gap), which was the original intent of the earlier figure. The new data regarding + and – RPA is important to document and we thank the reviewers for this constructive comment.

2. The use of the word “novel” in the title is not appropriate since the amino-terminus of RFC1 has been shown to bind DNA previously. This paper only clarifies how it binds and provides function. Words such as novel should be reserved for truly unique findings.

We agree, and have removed the term “novel” and changed the title to a more simple and direct explanation of what is observed in this report: **“**Cryo-EM structures reveal that RFC recognizes both the 3’- and 5’-DNA ends to load PCNA onto gaps for DNA repair”.

3. The observation of the ring closed form of RFC-ATPgammaS-DNA-PCNA is unexpected because it shows that closure can occur before hydrolysis and the main function of hydrolysis is the release of RFC. However, kinetic studies (e.g. a 2012 Hingorani JMB paper) show that with ATP, hydrolysis precedes closure and this should be clarified in the paper.

We clarify this in the Discussion as follows:

“Structural work of this study and others (Gaubitz et al., 2022) show that the clamp can close around dsDNA without ATP hydrolysis, while ensemble studies have indicated that stable clamp closure occurs after ATP hydrolysis (Anderson et al., 2009; Bowman et al., 2004; Liu et al., 2017; Marzahn et al., 2015; Sakato et al., 2012b; Trakselis et al., 2003). Both the structural and bulk results may be correct considering that “stable” closure of the clamp may require ATP dependent RFC dissociation.”

4. Biochemical studies with the isolated BRCT domain show that it binds partial double-stranded DNA with a 5’-phosphorylated junction with much higher affinity than the same DNA lacking the 5’-phosphate. The 1998 NAR Hardy study and the 2010 Siegal paper describing these studies have been referenced, but they were not discussed. Why were the cryoEM studies carried out with DNAs lacking the 5’-phosphate?

The reviewers have a good point. The main reason that most of our work did not use 5’ P is because neither of the earlier papers in 2003 that documented 5’ 911 loading by Rad24-RFC and RAD17-RFC used a 5’-P (or if they did we did not see this in the methods). We therefore played it safe by using the same 5’ OH end as the early biochemical studies. For example, it was possible that phosphatase action might occur in the cell to produce a 5’ OH before the RAD-RFC clamp loaders functioned. On hindsight we would have used a 5’ P.

The early biochemical studies on the isolated BRCT domain of RFC1 binding to 5’-P-DNA mentions a Kd value of 10 nM, but it does not document this, and instead says: “data not shown”. Thus we are unsure if binding measurements were correctly evaluated. But assuming the Kd measurements were correct, and to inform readers, we have now added the following in the Discussion part 3:

“We note that early studies using the isolated BRCT domain of RFC1 indicate a very tight affinity (K of 10 nM) to recessed 5ʹ phosphate DNA (Kobayashi et al., 2010; Kobayashi et al., 2006). Combined with the very different structure predicted for the 5’—DNA bound BRCT domain from NMR studies (Kobayashi et al., 2010), it is possible that the free BRCT domain has a unique structure that binds 5ʹ–P DNA tightly, and then changes its conformation upon binding to the shoulder of Rfc1. Consistent with this speculation, we see very little difference (i.e. about 2-fold) in binding to DNA of RFC and RFC 86 lacking the N-terminal BRCT region. Indeed, a conformation change that favors a lower affinity of RFC to DNA may facilitate the ejection of RFC from PCNA-DNA that is needed for PCNA to bind other factors.”

5. Can the orientation of the 5’-DNA molecule be defined as unambiguous in these structures? If so, and the 5’-OH is not engaged in interactions, what sets the orientation as 5’ at the BRCT domain?

Yes, the 5’ DNA orientation is unambiguous from the CryoEM resolution and bp identification as mentioned in the text.

As shown in the new Supplementary Figure 7, the 5’ OH is located inside the 5’-P binding pocket. The 5’-DNA2 orientation is not determined by the specific interaction between the 5’-P and the binding cavity of the Rfc1. It is instead determined by the exact location of the 5’-end (OH or P), which in turn specifies the exact locations of the major and minor grooves to which the Rfc1 binds. In other words, the whole extended binding groove for the 5’-DNA2, in particular the cavity location for the 5-end, sets the orientation of the shoulder DNA.

6. In Figure 3D, it appears that the 5’-junction does not contact specific residues in the BRCT. We need to see better views of these interactions. And the same for the structure with the open PCNA. This is the major point in the paper and it is not well presented. Interestingly, the model structure 2K7F from Siegal shows that the 5’-phosphate is engaged in binding positive residues, as perhaps could be expected from imposing a high binding energy for such an interaction in the odelling program. Could the authors comment on whether the chain direction for the second DNA molecule could be ascertained directly from the EM maps? Still, this entire, critical point deserves more attention than it has been given.

We thank the reviewers for this comment and have added a new figure (Supplemental Figure 7) to illustrate the BRCT interaction with the 5’ end and more specifically, the cavity surrounding the 5’ end of DNA2 and in comparison, with the 5’ end cavity in Rad24. This new description is added in the revised text as follows:

“The recessed 5ʹ end is in a cavity lined by three basic residues, Lys-208 of the BRCT domain and His-556 and His-659 of the Rfc1 collar domain. The cavity is larger than necessary for accommodating the 5ʹ-OH used in this study but fits the 5ʹphosphate snuggly with two potential H-bonds (Supplementary Figure 7a-b). The cavity for the 5ʹ-recessed end in the alternative clamp loader Rad24-RFC is very similar and is also lined by one Lys residue and two His residues (Supplementary Figure 7c).”

7. Two experiments have been included in supplementary 9 and 10, which indicate that the presence of phosphate does not affect DNA binding. But in these studies, the bulk of binding energy comes from the ATPgammaS-clamploader-PCNA interactions with DNA. What about the binding of RFC to DNA in the absence of ATP or ATPgammaS?

We appreciate this comment and have now examined RFC binding to a 10 mer gap DNA plus and minus nucleotide. The results are given in the new Supplementary Figure 10 and shows that ATPgS is required for RFC to bind to DNA. This conclusion is also consistent with our earlier published work (Johnson et al., J Biol Chem *281*, 35531-35543), showing that ADP does not support DNA binding by RFC, and this earlier work is now cited in the revised manuscript.

In response to this comment, we have included a Supplementary figure showing the data of the comparison of RFC binding to DNA in the presence or absence of ATPgS using a 10mer gapped DNA. (i.e. Supplemental Figure 10).

8. Biochemical analysis RFC-PCNA on DNA with nicks and varying length gaps.a. The authors conclude that RFC binds tightly to all the substrates tested but binds with a two-fold (~15 nM vs ~30 nM) higher affinity for DNA molecules with gaps that have five or more bases.b. The 5 nucleotide or large gaps compares well with the structural finding.c. The presence of PCNA in the measurement reveals tighter binding than in its absence, but with the same trend as seen when PCNA was omitted.d. Analysis of an RFC ensemble whose RFC1 lacks the BRCT domain revealed 2-fold tighter binding (16 nM vs. 32 nM) when BRCT was present than when not.In view of the extensive set of contacts to DNA mediated by the BRCT domain, it is surprising that there is not a greater difference in Kd between primer-template and gapped DNA structures. Likewise, why does deletion of BRCT hardly change the affinity for DNA?

We appreciate this question: “Why does deletion of the BRCT domain only change the affinity of RFC for DNA by 2-fold?” One possibility is the BRCT domain does not contribute very much to the binding affinity of RFC-5’ DNA. But prior literature (Kobayashi et al., 2006) reports that the BRCT domain binds 5’ DNA at around 10 nM, although this was reported as “data not shown” in the original report, so it is difficult to evaluate it. However, one can propose the following. In a later report (Kobayashi et al., 2010), the same group predicted a preliminary structure of the isolated BRCT domain using NMR data, and their proposed structure is very different from the BRCT domain in our cryoEM derived structure of RFC-DNA-PCNA in which the BRCT domain is not free, but is instead on the shoulder of Rfc1 with 5’ DNA bound. Thus, perhaps the isolated BRCT domain has a unique structure in solution, before it binds the shoulder, and this alternative BRCT structure might bind DNA tight (e.g. maybe to recruit it to RFC), followed by a conformation change and lower affinity upon binding the shoulder of RFC. In this connection, it is important RFC does not bind too tight to 5’ DNA, or else it would not easily eject from PCNA-DNA, required for other proteins to function with PCNA.

The modifications to the manuscript that address this are those described above in point #4.

9. The Kd values for RFC (-/+ PCNA) binding of various DNA structures (Figure 4, etc) were calculated from experiments where RFC was titrated against a DNA concentration held fixed at 10 nM. Given that the measured Kds values were in the ~15-40 nM range, might the fixed concentration of DNA be too high to reveal an accurate Kd? Might this explain why little or difference in Kd was found?If maintaining the fixed concentration of DNA at 10 nM is required by the experiment, other approaches to extracting (perhaps) more accurate Kds might be found here:Jarmoskaite, I., AlSadhan, I., Vaidyanathan, P. P. and Herschlag, D. How to measure and evaluate binding affinities. eLife 9, e57264 (2020).

As anticipated by the reviewers, we performed the work at the lowest DNA concentration (10 nM) at which we could obtain confident fluorescent readings, while we maintained the ability to cover an extensive range of concentration points (from 1/20 x to 24 x). We appreciate the Herschlag reference, and have typically used the alternative quadratic equation described in Herschlag (equation 5 in his paper), in our past studies. The instrument we used for fluorescence measurements of this report had the GraphPad single site equation, which is actually a quite high end analysis, but in response to the reviewer’s comments we purchased a software package OriginPro software (OriginLab Corporation, Northampton, MA, USA), and re-analyzed all our titration data using the single site quadratic equation as described in equation 5 of Hershlag’s paper (Jarmoskaite et al., eLife 2020) as is now described in the revised methods section of the revised paper. We have input all the binding data and have performed the alterative curve fitting and replaced the main text Figure 4, and the all the detailed titration plots in the Supp Figs, with the new curve fitting of the data. The results are not very different from our earlier submission, and do not change the conclusions, but perhaps the apparent KD values might be more accurate.